# Temporal pattern and synergy influence activity of ERK signaling pathways during L-LTP induction

**Nadiatou T Miningou Zobon[1], Joanna Jędrzejewska-Szmek[2], Kim T Blackwell[3,4]\***

[1]Department of Chemistry and Biochemistry, George Mason University, Fairfax, United States; [2]Laboratory of Neuroinformatic, Nencki Institute of Experimental Biology of Polish Academy of Sciences, Warsaw, Poland; [3]Interdisciplinary Program in Neuroscience, Bioengineering Department, George Mason University, Fairfax, United States; [4]Krasnow Institute for Advanced Study, George Mason University, Fairfax, United States

**Abstract** Long-lasting long-term potentiation (L-LTP) is a cellular mechanism of learning and memory storage. Studies have demonstrated a requirement for extracellular signal-regulated kinase (ERK) activation in L-LTP produced by a diversity of temporal stimulation patterns. Multiple signaling pathways converge to activate ERK, with different pathways being required for different stimulation patterns. To answer whether and how different temporal patterns select different signaling pathways for ERK activation, we developed a computational model of five signaling pathways (including two novel pathways) leading to ERK activation during L-LTP induction. We show that calcium and cAMP work synergistically to activate ERK and that stimuli given with large intertrial intervals activate more ERK than shorter intervals. Furthermore, these pathways contribute to different dynamics of ERK activation. These results suggest that signaling pathways with different temporal sensitivities facilitate ERK activation to diversity of temporal patterns.

**\*For correspondence:**
kblackw1@gmu.edu

**Competing interests:** The authors declare that no competing interests exist.

## Introduction

Temporal patterns are a key feature of the environment. The speed of traversing space is indicated by the time between spatial cues. In classical conditioning, adaptive changes in behavior require an animal to respond to a cue in an appropriate time frame to gain a reward or avoid punishment (*Delamater and Holland, 2008*; *Mauk and Ruiz, 1992*). In these and other tasks, environmental cues, i.e., sensory inputs, are converted into spatio-temporal patterns of activation. Pattern discrimination requires that neurons are able to discriminate and respond differently to these different temporal input patterns (*Bhalla, 2017*); conversely, neurons need to be able to learn *despite* a difference in temporal patterns.

Long-term potentiation (LTP), the activity-dependent and persistent strengthening of synapses, is widely considered to be a cellular mechanism underlying memory storage. The temporal pattern sensitivity of neurons is evident from the wide range of synaptic plasticity induction protocols, with some protocols producing LTP and some producing weakening of synapses (*Malenka and Bear, 2004*). Memory consolidation is also affected by temporal patterns: it has been shown that theta oscillations, which emerge during learning, and sharp-wave ripples can influence LTP induction (*Çalışkan and Stork, 2018*; *Sadowski et al., 2016*; *Wang et al., 2020*), demonstrating a role for temporal patterns during memory consolidation.

In the hippocampus, two 'phases' of LTP are distinguished. Late-phase LTP (L-LTP), induced by repetitive stimulation, can last more than 8 hr and involves de novo protein synthesis (*Davis et al., 2000*; *Sweatt, 2004*; *Tang and Yasuda, 2017*). In contrast, early-phase LTP does not require protein

synthesis and typically does not persist for more than 2 hr. It is also important to distinguish between induction and maintenance of L-LTP. Induction of L-LTP refers to the processes occurring during and shortly after the induction stimulation, whereas maintenance of L-LTP refers to the processes occurring tens of minutes to hours after induction.

A critical molecule in induction of L-LTP and memory is extracellular signal-regulated kinase (ERK) (*English and Sweatt, 1997*). Numerous studies have demonstrated that ERK contributes to L-LTP induction by controlling the transcription and protein translation necessary for L-LTP (for review, see *Miningou and Blackwell, 2020*; *Peng et al., 2010*), as well as increasing cell excitability by phosphorylating potassium channels (*Schrader et al., 2006*). ERK is activated by a cascade of kinases (*Buscà et al., 2016*; *Terrell and Morrison, 2019*), which are activated by Ras family GTP binding proteins, which are controlled by diverse Guanine nucleotide Exchange Factors (GEFs) and GTPase-activating proteins (GAPs). Several of these GEFs and GAPs play a pivotal role in synaptic plasticity and neurodevelopmental disorders, but their role in ERK activation during induction of L-LTP has not been demonstrated.

In the hippocampus, synaptic stimulation leads to activation of these GEFs and GAPs. Influx of calcium activates the calcium-sensitive GEF, RasGRF (*Feig, 2011*; *Schwechter et al., 2013*), and, via CaMKII, enhances activity of the synaptically localized GAP, SynGap, while dispersing it from the spine (*Gamache et al., 2020*). The intracellular second messenger cyclic adenosine monophosphate (cAMP) activates two key molecules upstream of ERK activation: protein kinase A (PKA) and the GEF exchange factor activated by cAMP (Epac) (*de Rooij et al., 2000*; *Enserink et al., 2002*; *Schmitt and Stork, 2002*). However, the contribution of calcium and cAMP pathways to ERK activation during L-LTP induction is unclear. A critical question is to what extent those signaling pathways work synergistically to activate ERK, whether they are redundant or whether they operate in different temporal regions.

A second critical question is how patterns of synaptic input determine which set of signaling pathways activate ERK leading to the induction of L-LTP. Key ERK activators, such as PKA and CaMKII, exhibit different temporal sensitivity: cAMP/PKA favors spaced LTP stimulation protocols (*Kim et al., 2010*; *Scharf et al., 2002*; *Woo et al., 2003*), while calcium/CaMKII prefers massed LTP stimulation (*Ajay and Bhalla, 2004*; *Kim et al., 2010*). Both of these signaling pathways converge on ERK, allowing the neuron to learn despite a variation in temporal pattern; on the other hand, temporal pattern can still influence the response of the neuron by controlling the temporal pattern of ERK activation. For example, transient ERK leads to proliferation, while sustained ERK leads to differentiation (*Santos et al., 2007*; *Sasagawa et al., 2005*; *von Kriegsheim et al., 2009*).

To answer these questions about temporal pattern and synergy, we developed a single-compartment computational biochemical model of postsynaptic signaling pathway underlying L-LTP induction in hippocampal CA1 pyramidal neurons. Simulations reveal that ERK pathways work synergistically, that the contribution of cAMP increases while the calcium contribution decreases with spaced stimuli, with ERK activity favoring spaced stimuli. The calcium- and cAMP-activated pathways have distinct temporal dynamics, providing a mechanism whereby different temporal patterns can activate different downstream effectors.

## Results

### Model validation

To investigate how temporal pattern of synaptic activity determines which pathway in the ERK signaling cascade (*Figure 1A*, *Figure 1—source data 1–5*) dominates in dendritic spines of hippocampal CA1 pyramidal neurons, we developed a single-compartment, stochastic reaction–diffusion model of pathways activating ERK. The model included two calcium signaling pathways (RasGRF and SynGap) and three cAMP signaling pathways (Epac, PKA, and the βγ subunit of Gi). The model was built by merging and adapting two existing models of synaptic plasticity in hippocampal neurons (*Jain and Bhalla, 2014*; *Jędrzejewska-Szmek et al., 2017*). These previously published models were modified by adding SynGap and RasGRF, which are critical for ERK activation.

To validate the model, we replicated the results from several published experiments. First, we simulated an experiment measuring RasGTP in response to glutamate uncaging at 0.5 Hz with 0 mM extracellular $Mg^{2+}$ (*Harvey et al., 2008*), by delivering a single pulse of 0.5 μM of calcium for 60 s.

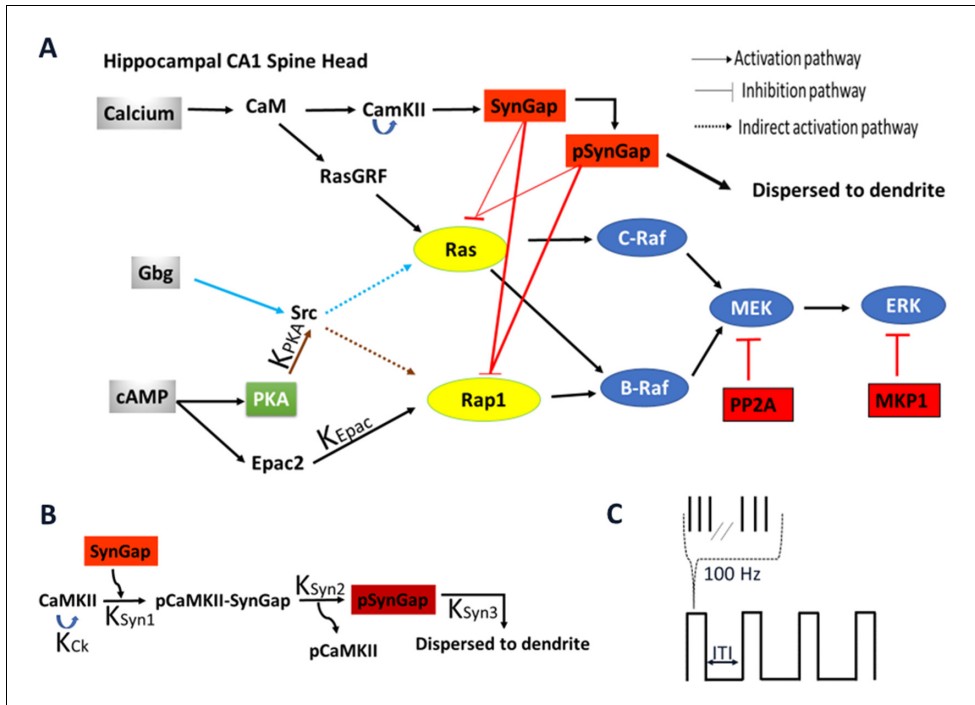

**Figure 1.** Schematic representation of signaling pathways activating ERK. (**A**) Five pathways are included in the model: calcium activation of (1) RasGRF followed by RasGTP production or (2) CaMKII phosphorylation of SynGap followed by increasing RasGTP and Rap1GTP lifetime; cAMP activation of (3) Epac or (4) PKA phosphorylation of Src family kinase, leading to Rap1GTP production; Gi subtype of GTP binding protein (Giβγ) (5) recruits Src family kinase followed by activation of RasGTP. (**B**) Four effects of CaMKII on SynGap were evaluated. $K_{CK}$: No autophosphorylation of CaMKII, $K_{Syn1}$: No pCaMKII binding to SynGap, $K_{Syn2}$: SynGap binds to pCaMKII, but cannot be phosphorylated, $K_{Syn3}$: SynGap phosphorylated, but not dispersed to the dendrite. (**C**) LTP protocols: four trains of 100 Hz (each 1 s duration) spaced by different intertrial intervals: 3, 20, 40, 80, and 300 s. The online version of this article includes the following source data for figure 1:

**Source data 1.** Reactions and rate constants involved in signaling pathway leading to Ras and Rap1 activation.
**Source data 2.** Reactions and rate constants involved in signaling pathway of SynGap.
**Source data 3.** Reactions and rate constants involved in core ERK signaling pathway.
**Source data 4.** Reactions and rate constants involved in signaling pathways from calcium to CaMKII.
**Source data 5.** Reactions and rate constants involved in signaling pathways leading from cAMP to PKA.
**Source data 6.** Total concentrations of molecule species.

RasGTP dynamics are qualitatively consistent with experimental measurements of *Harvey et al., 2008* (*Figure 2A*). Second, we mimicked an L-LTP experiment that applied two trains of 100 Hz frequency spaced by 20 s intervals (*Kasahara et al., 2001*), by injecting 5 µM calcium, 1 µM cAMP, and 0.1 µM Giβγ. The simulation shows qualitative match of ERK activation to the experimental model as ERK peaks around 3 min after stimulation and returns to basal about 30 min after stimulation (*Figure 2B*). Third, we show that cAMP-bound Epac and phosphorylation of PKA substrates are similar to that measured in hippocampal neurons (*Figure 2—figure supplement 1A and B*). Fourth, we show that our parameter optimization was successful, allowing us to reproduce CaMKII activation (*Figure 2—figure supplement 1C*) and CaMKII phosphorylation of SynGap (*Figure 2—figure supplement 1D*). Additional validations, comparing results to knockout experiments, are shown after the L-LTP results. In summary, the model is based on two, well-constrained, and validated signaling pathway models and is further validated experimentally with several systems level experiments.

## Single-pulse stimuli: maximal ERK activation requires multiple pathways

In order to assess whether multiple pathways are required for maximal ERK activation, both single pathway and multi-pathway dynamics were monitored. Simulation experiments used different

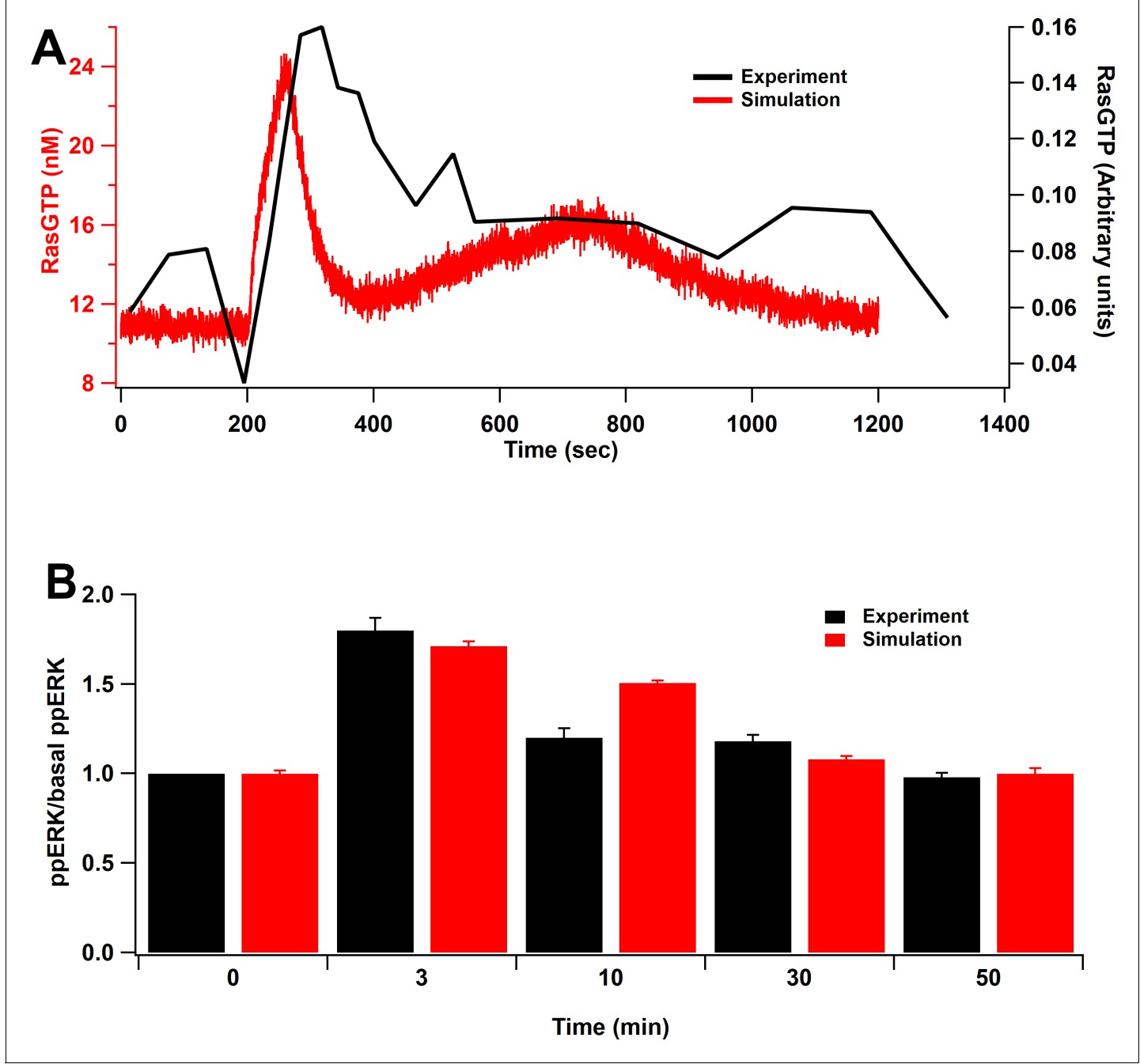

**Figure 2.** Model validation. (**A**) Time course of RasGTP in response to 30 pulses at 0.5 Hz (*Harvey et al., 2008*). RasGTP dynamics are qualitatively consistent with experimental measurements. (**B**) Relative increase of ppERK in response to L-LTP stimuli (two bursts of 100 Hz for 1 s, bursts separated by 20 s) (*Kasahara et al., 2001*). Simulation shows qualitative match of ERK activation to the experimental model as ppERK peaks around 3 min after stimulation and returns to basal about 30 min after stimulation.

The online version of this article includes the following figure supplement(s) for figure 2:

**Figure supplement 1.** Model validation.

amplitude ranges (for 1 s) and different durations of the input to evaluate doubly phosphorylated ERK (ppERK) (*Table 1*). We assessed whether the ERK response is graded or ultrasensitive and whether multiple pathways combine linearly or nonlinearly.

**Table 1.** Single pathway experiment.

| Input | Amplitude protocol (µM) | Duration protocol (s) |
|---|---|---|
| Calcium | 1 s: 0.2, 0.3, 0.5, 1, 1.5, 2, 5 | 0.5 µM: 1, 2, 4, 5, 6, 8, 10, 30, 100<br>1 µM: 1, 4, 100 |
| cAMP | 1 s: 0.1, 0.2, 0.5, 1, 1.5, 2 | 0.1 µM: 1, 4, 10, 30, 100<br>0.5 µM: 1, 4, 10, 30, 100 |
| Giβγ | 1 s: 0.005, 0.03, 0.1, 0.5 | 0.03 µM: 1, 4, 10, 30, 100<br>0.1 µM: 1, 4, 10, 30, 100 |

## Linear activation by cAMP pathway: opposite dynamics of Epac and PKA

cAMP activates ERK through three different pathways: Epac activation of Rap1GTP, PKA phosphorylation of Src, and PKA phosphorylation of β adrenergic receptors (βAR), which switches βAR coupling from Gs to Gi, which is followed by Giβγ recruitment of Src (*Lin et al., 2013*; *Luttrell et al., 1997*). To separate the direct action of cAMP (i.e. Epac activation of Rap1GTP or PKA phosphorylation of Src) from its indirect action through the switching pathway, we evaluated the effect of Giβγ separately and in combination with Epac and PKA direct activation.

Model simulations show a linear activation of ppERK with either cAMP duration or concentration (*Figure 3*, *Table 2*), through the direct pathway and a slightly non-linear activation with indirect pathway. As observed in *Figure 3A,B*, as the input duration increases, ppERK amplitude and duration increase. Similarly, the indirect cAMP pathway displays linear activation of ERK with Giβγ input (*Figure 3C*). *Figure 3D* shows that increasing the input duration (red traces) or concentration (black trace) is similarly effective in increasing ppERK. Giβγ produces a delayed activation of ERK consistent with experimental data (*Koch et al., 1994*; *Schmitt and Stork, 2002*) and a computational model showing a 7 min delay (*Khalilimeybodi et al., 2018*). cAMP and Giβγ input for these simulations are shown in *Figure 3—figure supplement 1*. In summary, in response to cAMP input, ppERK total activity is linearly related to the input duration and concentration, with a delayed activation through the indirect pathway.

To investigate the activation of ERK by Epac and PKA individually, the simulations were repeated while blocking cAMP binding to one of these molecules (*Table 3*). When cAMP binding to Epac is blocked, the basal quantity of ppERK is dramatically reduced (results not shown), implying that Epac is the main source of basal ppERK. The remaining activity, due to PKA, shows a delayed activation of ERK (*Figure 3F*). In contrast, Epac produces a fast and transient activation of ERK (*Figure 3F*). Similarly to PKA activation, Giβγ shows a delayed activation of ERK (*Figure 3F*). Thus, Epac contributes to ERK activation with different temporal dynamics compared to PKA and Giβγ.

Direct and indirect cAMP pathways combine linearly for short duration inputs and sublinearly with prolonged duration inputs. *Figure 3E* shows that the response to stimulating all three cAMP pathways overlaps with the sum of the single pathway responses only at a short duration. As the stimulus duration increases, the ppERK response is slightly lower than the sum of responses to cAMP direct and indirect activation (*Figure 3E,F*). Sublinear summation is due to two loci of competition. First, competition was observed between the PKA phosphorylation of Src and Giβγ activation of Src. PKA has a higher affinity for Src than does Giβγ; therefore, Src is trapped by PKA, reducing available Src for Giβγ. This sublinear response is observed with 2× and 4× increases in Src quantity (results not shown), demonstrating that the sublinear summation is not due to Src depletion but indeed due to Giβγ and PKA competition with Src. The second competition was observed between Epac and the Giβγ/Crk-C3G for Rap1GDP. Rap1GDP is utilized by Epac, which reduces the Rap1GDP availability for Giβγ/Crk-C3G. Blocking Epac allows more Rap1GDP to bind Crk-C3G, prolonging ppERK activity as observed with Giβγ alone. In conclusion, competition between PKA and Giβγ for Src, and between Epac and Crk-C3G for Rap1GDP, affects summation in the cAMP pathway.

## Non-linear activation by calcium pathways: SynGAP dispersion and phosphatase activity control ERK activation by calcium

Model simulations show a non-linear activation with either calcium duration or concentration. *Figure 4A* reveals that low calcium inputs produce little to no ppERK, whereas with higher calcium

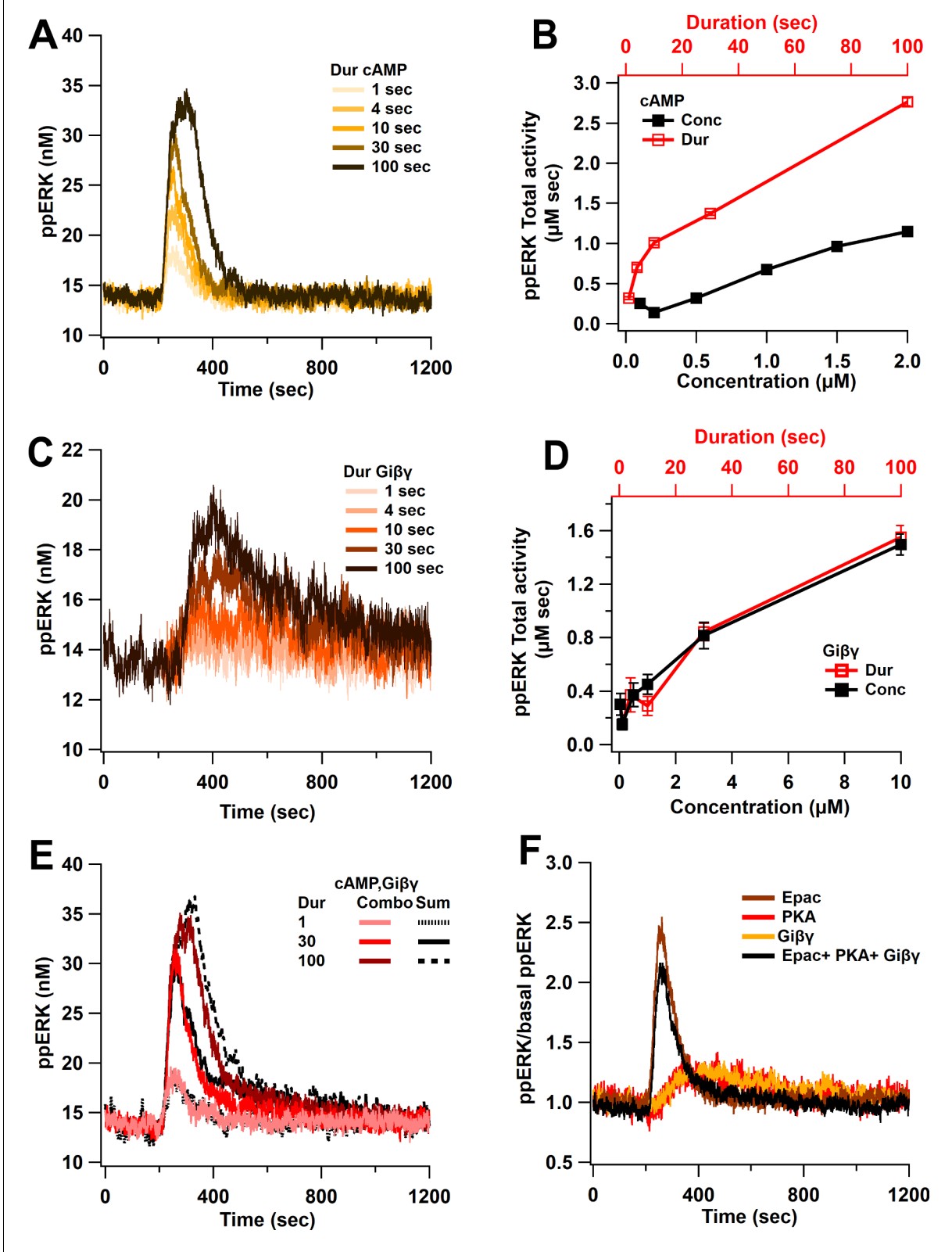

**Figure 3.** Linear activation of ERK by cAMP pathways. (A) ppERK response to a single pulse of 0.5 µM cAMP is graded with duration. (B) Integral of ppERK concentration over time (AUC) in response to different cAMP concentrations (for 1 s) and different cAMP durations (with a concentration of 0.5 µM). (C) Time course of ppERK in response to single pulses of 0.1 µM Giβγ for different durations. Recruitment of ERK by Giβγ binding protein shows delayed activation. (D) Total activity of ppERK in response to different Giβγ concentrations (for 1 s) and durations (with a concentration of 0.1 µM).

*Figure 3 continued on next page*

Figure 3 continued

Increasing duration or concentration are similarly effective in increasing ppERK. (E) Direct and indirect cAMP pathways combine linearly for all duration inputs and sublinearly, with prolonged duration inputs. Two-way ANCOVA (analysis of covariance) of ppERK AUC versus stimulation characteristics (duration) and type (combination versus summation) as factors (N = 5 in each group) is significant for both duration and type (F (2,47) = 419.5, T < 0.001; P(dur) < 0.0001, T(dur) = 28.61, P(type) < 0.0001, T(type) = 4.54). Combo: response to Giβγ and cAMP, sum: sum of the responses in (A) and (C). (F) Ratio of ppERK to basal ppERK in response to cAMP activation of PKA, Epac or Giβγ (0.5 µM (cAMP) or 0.1 µM (Giβγ) for 30 s). ppERK increases quickly and transiently with Epac and with a delay and more prolonged in response to PKA or Giβγ.

The online version of this article includes the following figure supplement(s) for figure 3:

**Figure supplement 1.** Time course of synaptic inputs for single-pulse simulations.

inputs, ppERK reaches a plateau and the duration of ppERK is prolonged. Similar results are observed with increasing duration using 0.5 µM calcium concentration. Total activity of ppERK (*Figure 4B*, *Table 2*) reveals a supralinear response (also called ultrasensitivity), with a calcium duration threshold of 5 s and an amplitude threshold between 1 and 2 µM. A calcium amplitude or duration above these thresholds induces maximal and sustained ppERK. Calcium input for these simulations is shown in *Figure 3—figure supplement 1A*.

To determine whether the ultrasensitivity of ppERK was caused by CaMKII, we blocked CaMKII phosphorylation (of itself and SynGap) and assessed ERK activation by RasGRF. Blocking CaMKII significantly reduces ppERK activity, especially for long or high calcium inputs (*Figure 4C*). We also observed a shift from a sustained and prolonged ppERK to a transient ppERK, which is caused by calmodulin activation of RasGRF (*Figure 4G*). This shows that RasGRF is responsible for the early calcium activation of ERK.

The ultrasensitivity of CaMKII autophosphorylation has been shown to depend on the quantity of the phosphatase PP1 (*Bradshaw et al., 2003*; *Singh and Bhalla, 2018*), and thus might regulate the calcium duration and amplitude thresholds. We used a concentration of 3.6 µM of PP1 (PP1: CaMKII ratio = 18%) based on previous reports (*Rangamani et al., 2016*). To evaluate whether PP1 quantity controls the ppERK non-linear response, we repeated simulations with different quantities of PP1 (10, 50, and 100% of CaMKII quantity). Model simulations show that both ppERK and phosphorylated CaMKII decrease as PP1 concentration increases (*Figure 4D*). The calcium duration producing half-maximal ppERK did not change much with increased PP1; however, the *sensitivity* to calcium duration decreased with an increase in PP1 (*Figure 4E*, *Table 4*). Thus, our data demonstrate that PP1 indirectly influences ppERK amplitude and sensitivity to calcium duration through modulation of pCaMKII. Therefore, the ultrasensitivity of both CaMKII and ppERK depends on the ratio of CaMKII to PP1 quantity.

CaMKII does not directly activate ERK but instead modulates ERK phosphorylation through its phosphorylation of SynGap, a molecule that inactivates Ras family GTPases. The effect of CaMKII phosphorylation is complex, including enhanced SynGap activity (*Walkup et al., 2015*) and SynGap dispersion from the spine (*Araki et al., 2015*). We evaluated the contribution of these processes by

**Table 2.** Regression analysis results for cAMP pathway.

AIC: Akaike information criteria, lower values indicate better models. If adjusted $R^2$ for linear is less than 0.9, then the best model is determined by AIC.

| | | Giβγ | | cAMP | | Calcium | |
|---|---|---|---|---|---|---|---|
| | | **Dur** | **Conc** | **Dur** | **Conc** | **Dur** | **Conc** |
| Adjusted $R^2$ | Linear | 0.832 | 0.827 | 0.951 | 0.937 | 0.486 | 0.675 |
| | log | 0.880 | 0.787 | 0.957 | 0.815 | 0.955 | 0.604 |
| | Hill | 0.922 | 0.913 | 0.969 | 0.964 | 0.970 | 0.989 |
| AIC | Linear | −1.112 | −8.266 | −10.181 | −52.828 | 166.225 | 99.958 |
| | log | 10.196 | 24.022 | 13.458 | 13.710 | 107.598 | 127.772 |
| | Hill | −28.406 | −38.033 | −48.562 | −48.911 | −86.367 | −101.415 |
| Conclusion | | Non-linear | Non-linear | Linear | Linear | Non-linear | Non-linear |

**Table 3.** Knockout experiment.

Rate constant names are shown in *Figure 1A,B*. Block lists the rate constant whose value is set to zero.

| Input | Block | Effect/molecular dependence |
|---|---|---|
| Calcium 2000 nM for 1 s | $K_{CK}$ | CaMKII is activated but cannot autophosphorylate itself, thus RasGRF is the only contributor |
| | $K_{Syn1}$ | pCaMKII is unable to bind to SynGap |
| | $K_{Syn2}$ | pCaMKII binds to SynGap but is unable to phosphorylate SynGap |
| | $K_{Syn3}$ | SynGap is phosphorylated but is unable to disperse to the dendrite |
| cAMP 500 nM for 30 s | $K_{Epac}$ | Epac binds cAMP but cannot activate Rap1GDP, thus PKA is the only contributor |
| | $K_{PKA}$ | Final step of PKA pathway activation of Rap1GDP is blocked, thus Epac is the only contributor |

eliminating them individually. Simulations reveal that persistent activation of ERK requires phosphorylation of SynGap by pCaMKII and dispersion of pSynGap to the dendrite (*Figure 4F*). A lower ppERK total activity results when pSynGap remains anchored in the spine (*Figure 1B*, *Figure 4F*, no pSynGap dispersion trace). Blocking the phosphorylation of SynGap was implemented using two approaches. In one, CaMKII can bind to SynGap but does not phosphorylate it. This reduces the available SynGap, which doubles ppERK concentration compared to control (*Figure 4F*, pCK bound to SynGap). In the second approach, CaMKII is unable to bind to SynGap. This increases the available SynGap, resulting in a reduction of ppERK below basal (*Figure 4F*, no pCK bound to SynGap). These data suggest that the quantity of free SynGap in the spine, which is controlled by CaMKII, regulates ERK activation. To verify that the results were not an artifact of the binding affinity implemented in the model, simulations were repeated using 10× and 100× lower affinity of pCaMKII for SynGap. Simulation results show a similar relation of ppERK to pSynGap, reinforcing the hypothesis that CaMKII regulates ppERK total activity through its modulation of SynGap.

## Overall ppERK response is a linear combination of response to calcium and cAMP

Simulations reveal that both pathways (calcium, cAMP) combine linearly to activate ERK. *Figure 5A* shows that the response to stimulating both pathways overlaps with the sum of the single pathway responses. This summation was observed with short and long duration stimuli and with high- and low-amplitude stimuli, regardless of whether calcium was below or above the threshold. When calcium is below threshold (e.g. short duration stimuli), cAMP contributes more to ppERK, whereas when calcium is above the threshold, it provides most of the ppERK (*Figure 5B*). The contribution of cAMP is transient, occurring briefly after induction while the calcium contribution is delayed and sustained (*Figure 5C*). The two pathways have distinct temporal dynamics that overlap and switch in their contribution about 3 min after induction. Further subdividing pathway contributions (*Figure 5D*) reveals that the transient ppERK is produced by Epac, with PKA and Gi contributing to a small sustained ppERK. Most of the sustained ppERK is produced by SynGap, with RasGRF contributing to a small transient ppERK.

In summary, ppERK exhibits a non-linear response to calcium, a linear response to cAMP pathways, and the response to the combination of pathways also is a linear combination of the single pathway responses. Nevertheless, each pathway exhibits distinct temporal pattern dynamics. However, these simulations used single pulses of stimuli; thus, the next section evaluates the response to stimuli similar to those used for synaptic plasticity experiments.

## L-LTP stimuli: ERK signaling pathway temporally integrates input signals

Previous research has shown that different temporal patterns for inducing L-LTP work through different signaling pathways: PKA is needed for spaced stimuli, while CaMKII is greater in response to massed stimuli. Thus, a critical question is whether different temporal patterns select different signaling pathways for activation of ERK. To evaluate that question, we simulated the response to L-LTP inputs. As in L-LTP experiments, the model was activated with four trains of 100 Hz stimuli for 1 s using a range of intertrain intervals (ITIs), including massed (3, 20, 40 s) and spaced (80,300 s) ITI.

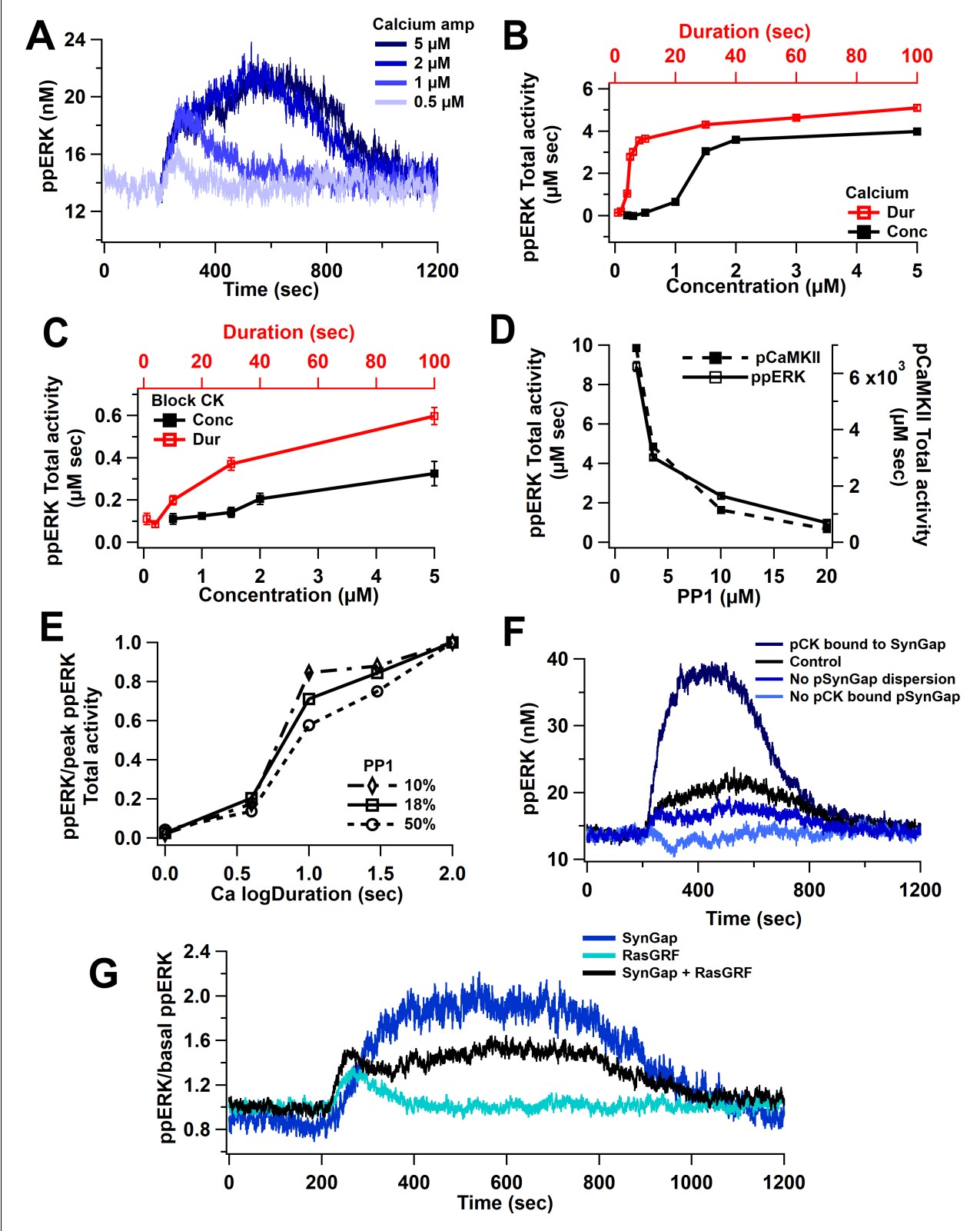

**Figure 4.** Nonlinear activation of ERK by calcium pathway. (**A**) Time course shows a sustained ppERK when calcium is above amplitude threshold (2 µM) for 1 s duration. (**B**) Total ppERK activity over time in response to different calcium concentrations (for 1 s) and different calcium durations (with a concentration of 0.5 µM) reveals ultrasensitive response. (**C**) With CaMKII activity blocked, RasGRF is the sole calcium-activated pathway to ERK activation. ERK activation is no longer ultrasensitive. (**D**) pCaMKII and ppERK activities in response to 0.5 µM for 30 s of calcium decrease with increased

*Figure 4 continued on next page*

Figure 4 continued

PP1. Similar results were observed with different concentrations or duration. (E) Increasing PP1 quantity decreases the steepness of ppERK vs duration (log10) curve. (F) Total ERK activity changes with SynGap availability. Blocking dispersion of SynGAP reduces ppERK, whereas allowing CaMKII to bind but not phosphorylate SynGAP enhances ppERK. (G) Ratio of ppERK to basal ppERK in response to calcium activation of RasGRF or SynGap alone (0.5 μM calcium for 30 s), compared with control (SynGap + RasGRF).

## Temporal sensitivity of ppERK to cAMP matches that of PKA

Since ppERK increases more with cAMP duration than concentration (either direct action or in combination with Giβγ), as shown above, we predicted that ppERK would be greater for spaced than massed stimulation with cAMP. Model simulations with cAMP input show that ppERK total activity increases as ITI increases (*Figure 6*). Note that the ppERK amplitude decreases with ITI (*Figure 6A*), but the increase in ppERK duration more than compensates for the lower amplitude, resulting in greater total activity for the 300 s ITI. The summation is significantly higher than the combination as observed with single-pulse stimuli: ppERK in response to Giβγ plus Epac and PKA pathways is less than the sum of the responses to either pathway alone (*Figure 6B*). Competition for Src and Rap1GDP, which limits linear summation, is also observed for L-LTP stimuli. Note that the temporal sensitivity of ppERK is similar to that of PKA (*Figure 6B*). However, the Epac pathway contributes more to ppERK with short ITI while PKA contributes more with spaced stimuli (*Figure 6—figure supplement 1*), suggesting an Epac-dependent L-LTP for short ITIs. In summary, ERK activated by cAMP exhibits sensitivity to ITI and is greater with spaced stimuli. These data are consistent with experiments showing a requirement for PKA in spaced but not massed stimulation, and simulations showing an increase in PKA activity with spaced stimulation (*Kim et al., 2010*; *Woo et al., 2003*).

## ppERK favors longer ITIs, independent of CaMKII temporal sensitivity

L-LTP simulations using 1 μM amplitude calcium pulses produce similar ppERK traces for ITIs between 3 and 40 s (*Figure 7A*). Multiple 1 s pulses of 1 μM calcium are similar to a single 5 s pulse, due to the slow decay of pCaMKII; thus, calcium is above the threshold for CaMKII ultrasensitivity and ppERK is sustained for a long duration, consistent with experimental data showing that CaMKII above threshold is enough to induce L-LTP (*Shibata et al., 2021*). Only with a 300 s ITI does pCaMKII and ppERK decay (though not to basal) between stimuli. The temporal sensitivity of ppERK in response to calcium input tends toward longer ITIs than the temporal sensitivity of CaMKII. The greatest total activity of ppERK (*Figure 7B*) and pCaMKII (*Figure 7C*) are observed with 80 s and 3 s ITI, respectively. To test the role of CaMKII ultrasensitivity in the temporal sensitivity of ppERK, we repeated simulations with an increased PP1 quantity (10 μM or 50% ratio). As observed with 18% PP1, the greatest pCaMKII occurs with short ITIs, as reported previously (*Ajay and Bhalla, 2004*; *Kim et al., 2010*), but the decrease in pCaMKII with ITI is steeper with 50% PP1. This change shifted ppERK temporal sensitivity slightly to favor middle ITIs but still does not match the temporal sensitivity of CaMKII.

In addition to varying PP1 quantity, we used lower concentration calcium pulses and an alternative CaMKII model (alt CaMKII) (*Dupont et al., 2003*; *Jędrzejewska-Szmek et al., 2017*) that does not exhibit the same ultrasensitivity. Simulations with this alternative CaMKII model reveal a small shift in temporal sensitivity toward smaller ITIs, but the temporal sensitivity of ppERK still tends

**Table 4.** PP1 concentration influences ppERK quantity and ultrasensitivity.

The calcium duration producing half-maximal ppERK AUC and the sensitivity to duration were estimated by fitting a hill equation to ppERK AUC vs calcium duration. Values shown are parameter estimates ± standard deviation of the estimate. Width at half max: the calcium duration at half-maximal ppERK.

| Pp1 (%) | Width at half max | Hill coefficient |
| --- | --- | --- |
| 10 | 6.0 ± 1.07 | 4.1 ± 1.5 |
| 18 | 6.7 ± 1.53 | 2.5 ± 1.1 |
| 50 | 9.9 ± 5.28 | 1.4 ± 1.1 |

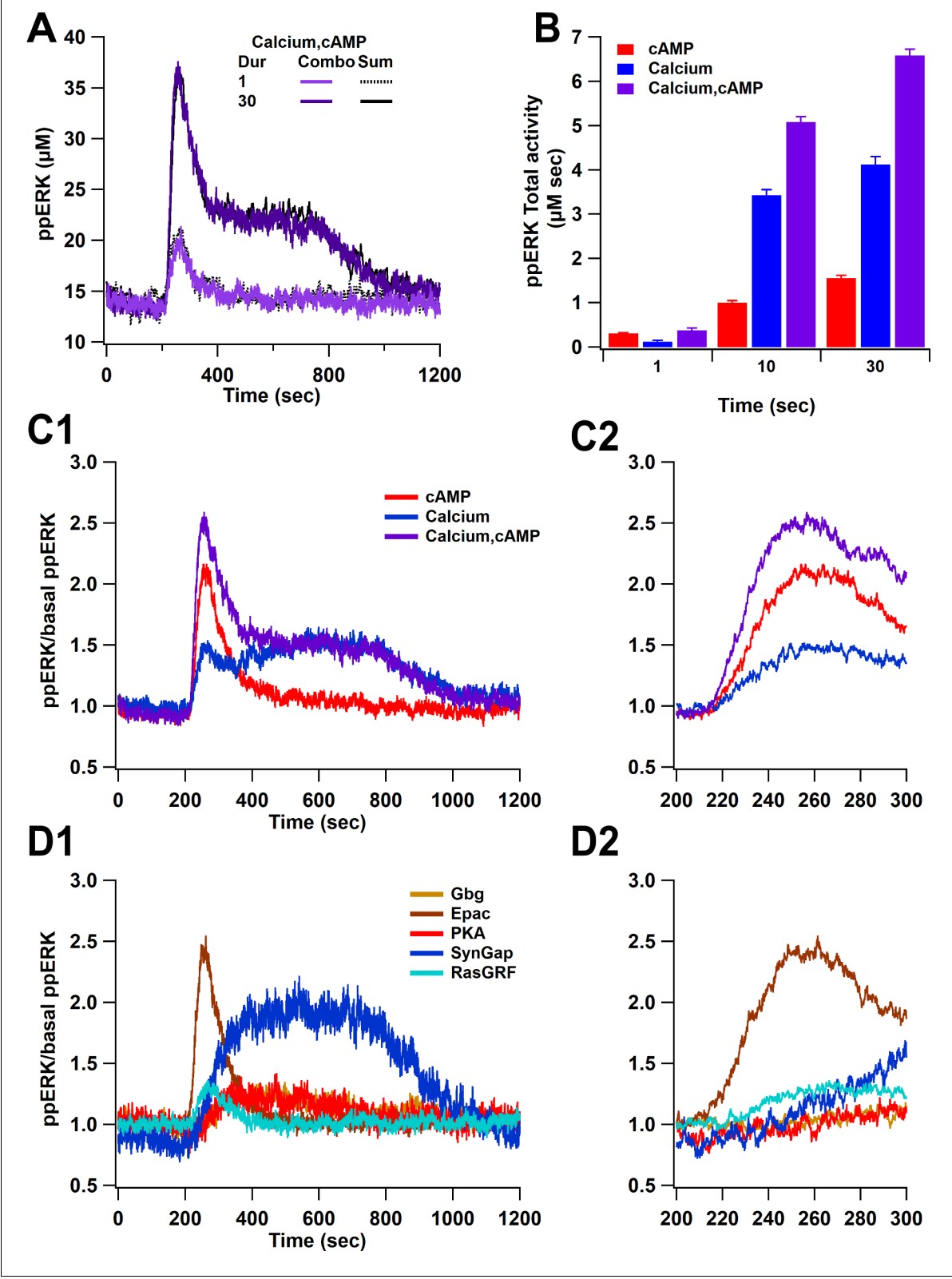

**Figure 5.** Overall ppERK response is a linear combination of response to calcium and cAMP. (**A**) Time course of ppERK response to calcium and cAMP. Calcium in combination with cAMP traces overlaps with the sum of the single pathway responses for both short and long duration stimuli. Two-way ANCOVA (analysis of covariance) of ppERK AUC versus stimulation characteristics (duration) and type (combination versus summation) as factors (N=5 in each group) is significant for both stimulation but not type (F (2,47) = 65.55, P(dur) < 0.0001, T(stim) = 11.41, P(type) = 0.35, T(type) = −0.945). (**B**)

*Figure 5 continued on next page*

*Figure 5 continued*

Summary of ppERK AUC versus duration shows that the fraction of ppERK produced by calcium pathways increases with duration of the stimulation. (C) Dynamics of ppERK, relative to basal ppERK, in response to cAMP (0.5 μM cAMP and 0.1 μM Giβγ for 30 s), calcium (0.5 μM for 30 s), or combination. ppERK increases transiently with cAMP, with a delay in response to calcium and is higher with combination (calcium and cAMP ). (D) Dynamics of single pathways. Ratio of ppERK to basal ppERK in response to PKA (0.5 μM cAMP), Epac (0.5 μM cAMP), Giβγ (0.1 μM), RasGRF (0.5 μM calcium), and SynGap (pCaMKII feedforward loop; 0.5 μM calcium) for 30 s. Epac and RasGRF produce transient ppERK, whereas PKA, Giβγ, and SynGap produce transient and delayed ppERK. C2 and D2 expand the first 100 s after stimulation of C1 and D1, respectively.

toward longer ITIs than the temporal sensitivity of CaMKII. Similar results were observed with the ultrasensitive CaMKII model when smaller calcium pulses (0.5 μM) were used (*Figure 7B,C*). In addition, ERK's temporal sensitivity follows RasGRF (low calcium) and SynGap (high calcium) temporal

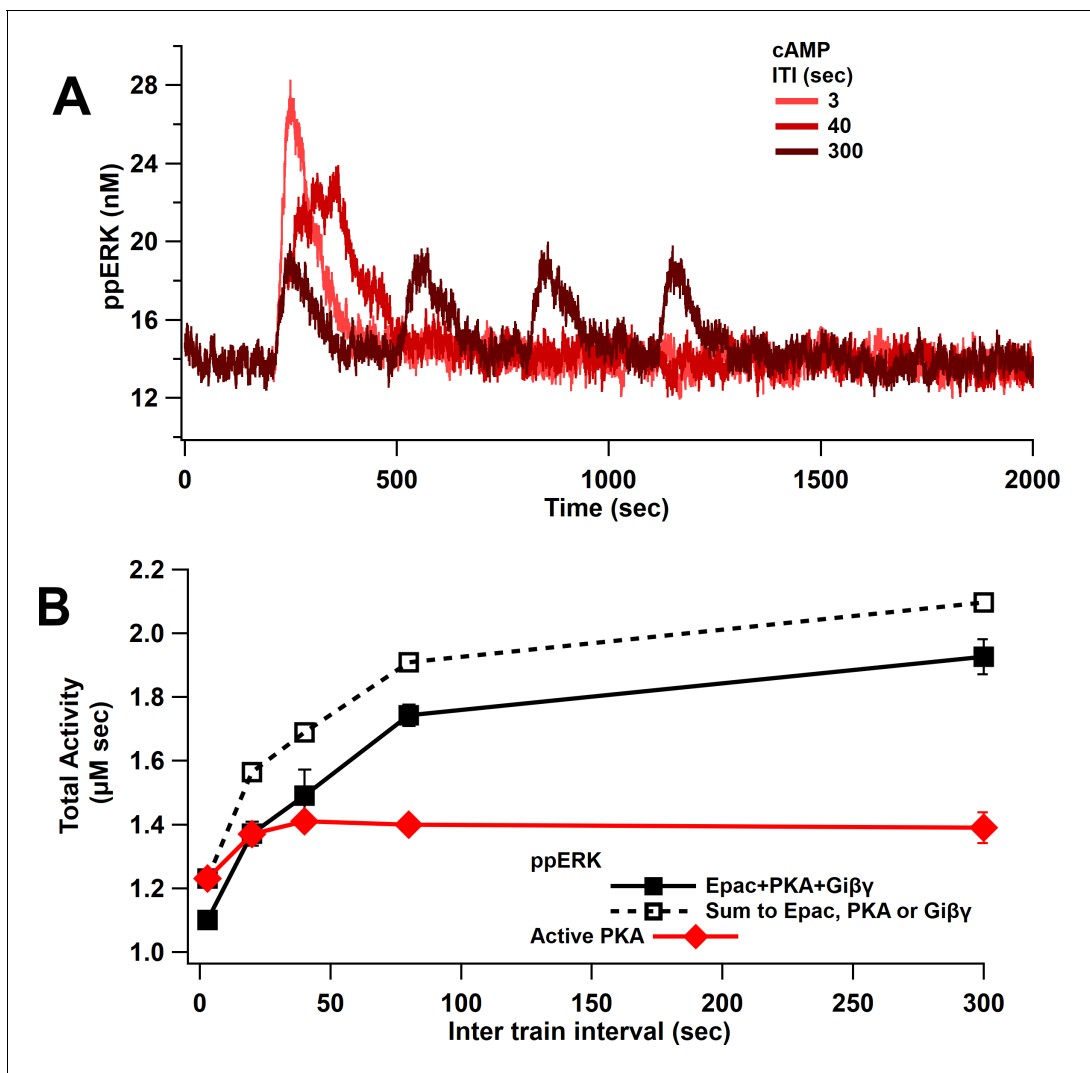

**Figure 6.** ppERK favors spaced stimuli in response to cAMP input. (A) Time course of ppERK in response to 4 trains of 100 Hz stimuli of cAMP (Epac, PKA, Giβγ pathways). ppERK peak amplitude decreases with ITI, but the AUC increases with ITI. (B) Total kinase activity in response to cAMP shows that ppERK exhibits sublinear response to cAMP stimuli: the sum (dashed line) is higher than ppERK in response to combination (solid black line). PKA and ppERK have similar temporal sensitivity, favoring spaced stimuli. Two-way ANCOVA (analysis of covariance) of ppERK AUC versus stimulation characteristics (ITI) and type (combination versus summation) as factors (N = 5 in each group) is significant for both stimulation ITI and type (F (2,47) = 42.25, T < 0.0001; T(ITI) = 4.67, P(ITI) < 0.0001, T(type) = 7.92, P(type) < 0.0001).

The online version of this article includes the following figure supplement(s) for figure 6:

**Figure supplement 1.** Uniform ppERK temporal sensitivity within cAMP pathways.

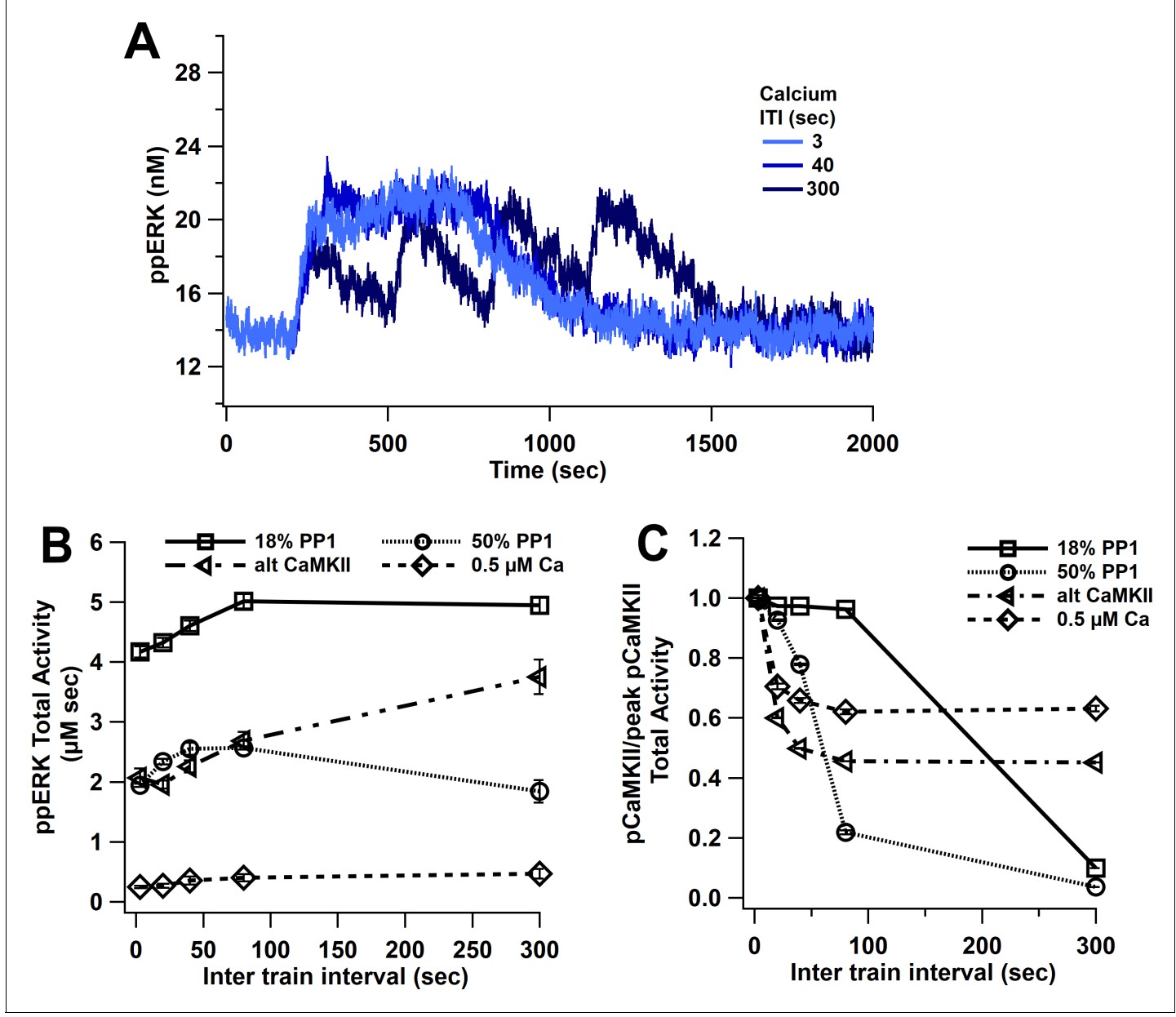

**Figure 7.** ppERK favors spaced stimuli in response to calcium input, in contrast to temporal sensitivity of CaMKII. (**A**) Time course of ppERK in response to four trains of 100 Hz calcium stimuli (1 μM concentration) shows sustained and prolong ppERK with 3 and 40 s ITI. (**B**) ppERK AUC changes with PP1 quantity, but still is highest for spaced ITIs. (**C**) Increasing PP1 quantity or lowering calcium concentration makes CaMKII more sensitive to shorter ITIs.

The online version of this article includes the following figure supplement(s) for figure 7:

**Figure supplement 1.** ppERK temporal sensitivity is similar to that of inactive SynGap.

**Figure supplement 2.** SynGap availability controls the magnitude of ppERK during L-LTP induction.

activity (*Figure 7—figure supplement 1*). Similar to what was observed with single-pulse experiments, either blocking dispersion of SynGap or blocking pCaMKII binding to SynGap reduces ppERK (*Figure 7—figure supplement 2*). In summary, though the ultrasensitivity of CaMKII is controlled in part by PP1, ppERK temporal sensitivity does not follow CaMKII temporal sensitivity, and instead follows the temporal sensitivity of CaMKII downstream targets such as SynGap.

## ppERK favors long intertrial intervals

To simulate L-LTP in response to four trains of synaptic activation, we used both calcium and cAMP inputs, as would occur with glutamatergic inputs. L-LTP simulations with both calcium and cAMP show an increase of ppERK total activity with longer ITI. Results show that the greatest total ppERK occurs at 80 s (using 18% PP1 and 1 µM calcium) or 300 s (using 50% PP1-1 µM calcium or 18% PP1-0.5 µM calcium) (*Figure 8A*). Though the change in optimal ITI varies little, the total quantity of ppERK greatly depends on calcium and PP1 concentration. In these simulations, the contribution of the cAMP pathway to ppERK is independent of calcium, in part because cAMP, calcium, and Giβγ are specified independently (*Figure 8—figure supplement 1*). To address the question of which pathway contributes more to ppERK and whether different temporal patterns use different pathways, we calculated the percent ppERK produced by each pathway. cAMP is responsible for a majority of the ppERK with below threshold calcium, but only responsible for a third of the ppERK for above threshold calcium (*Figure 8B*). Analyzing the five pathways separately reveals ppERK increases with ITI for all pathways (*Figure 8—figure supplement 2*). Nonetheless, the relative contribution of SynGap decreases with ITI, as the other pathways increase more with ITI (*Figure 8C*). As seen with single-pulse stimuli, pSynGap is the main source of prolonged ppERK, whereas Epac and RasGRF are main sources of transient ppERK (*Figure 8—figure supplement 3*).

ppERK linearly integrates the calcium and cAMP inputs when calcium is below threshold. Thus, the response to the combined stimuli is similar to the sum of ppERK responses to individual stimuli with 0.5 µM calcium (*Figure 8A*). Similar results were obtained with 50% PP1. With 18% PP1 and high calcium (above threshold), supralinear summation was observed: the response to the combination is greater than the summation of individual responses (*Figure 8A*). CaMKII modulation of SynGap with above threshold calcium may be responsible for the supralinear summation as SynGap dispersion enhances RasGTP and Rap1GTP in response to cAMP.

To further investigate responses to synaptic input, we simulated our model with five different L-LTP protocols. Two of them, bath-applied isoproterenol (ISO) followed by either 1 s of 100 Hz stimulation or 180 s of 5 Hz, experimentally elicit LTP, whereas giving 100 Hz, 5 Hz, or ISO alone do not. For these protocols, cAMP, calcium, and Giβγ inputs (*Figure 9—figure supplement 1*) were determined from a spatial model response to glutamate (*Jędrzejewska-Szmek et al., 2017*); thus, cAMP is produced by Gs and calcium-calmodulin activation of adenylyl cyclase. Simulation results show that isoproterenol followed by either 100 Hz or 5 Hz produce greater ppERK than ISO, 100 Hz, or 5 Hz alone (*Figure 9*). These results are consistent with experimental observations (*Gelinas et al., 2008a*; *Gelinas et al., 2008b*; *Winder et al., 1999*) that the combination stimuli are needed to produce L-LTP, thus validating the model. In addition, our results show that cAMP provided by ISO is compensating for the strong calcium input provided by four trains of 100 Hz.

We performed additional simulations to evaluate the role of Ras and Raf isoforms and to further validate the model. First, we simulated Raf knockout experiments and measured ERK activity in response to four trains of 100 Hz. Simulation results are consistent with experimental data (*Chen et al., 2006*; *Li et al., 2016*; *Takahashi et al., 2017*; *York et al., 1998*) as bRaf knockout, but not Raf1 knockout, greatly reduces ERK activity (*Figure 8—figure supplement 4A*). Second, we simulated Ras or Rap knockout and measured ERK activity after LTP induction. Simulation results are consistent with experimental data (*Grewal et al., 2000b*; *Grewal et al., 2000a*; *Keyes et al., 2020*; *Li et al., 2016*; *Takahashi et al., 2013*; *York et al., 1998*; *Zhang et al., 2018*) as knocking out Rap1 produces a greater ppERK deficit than knocking out Ras (*Figure 8—figure supplement 4B*). We assessed whether Ras could compensate for Rap and vice versa, by doubling the quantity of the remaining GTPase. Result shows that doubling Ras cannot compensate for the Rap1 knockout, whereas doubling Rap1 increases ppERK above the control (*Figure 8—figure supplement 4B2*). This shows, for the first time, that overexpression of Rap1 can compensate for Ras activity in ERK activation.

A unique aspect of our model is that full activation of Raf1-RasGTP requires dimerization; thus, we evaluated the impact of dimerization on ERK activation. Dimerization reduces the temporal sensitivity of ppERK (*Figure 8—figure supplement 4A*), as the difference in ppERK between 3 s and 300 s ITIs is much larger when dimerization does not occur. The biggest effect is a reduction in ppERK at short ITIs; thus, dimerization enhances ppERK mostly for short ITIs.

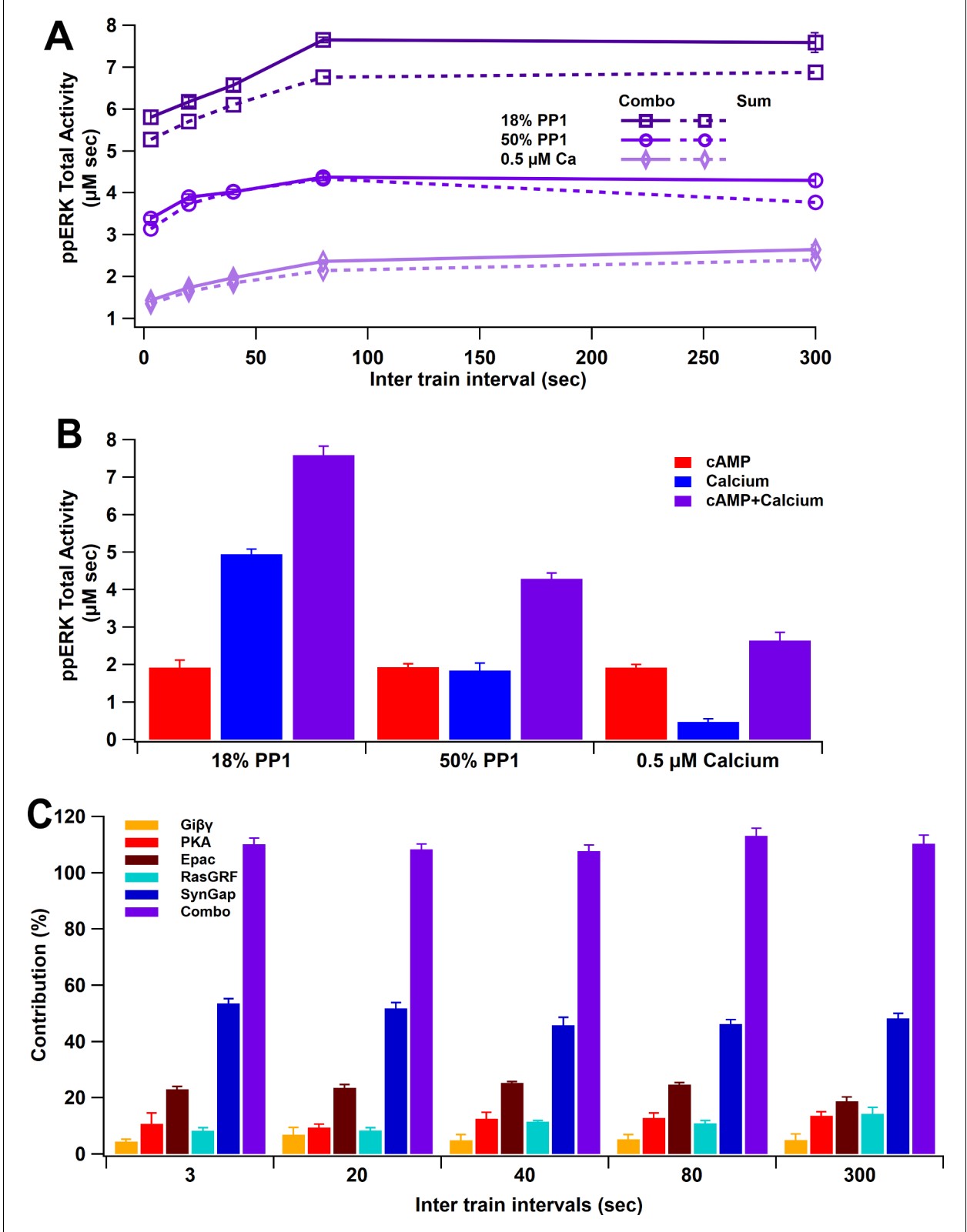

**Figure 8.** ppERK integration of calcium and cAMP inputs can be linear or supralinear. (**A**) ppERK mostly increases with temporal interval, with the greatest quantity occurring at 80 s (above calcium threshold or 18% PP1) or 300 s (below calcium threshold). Supralinear summation occurs at 18% PP1 and linear summation occurs at low calcium or high PP1 – conditions that reduce ultrasensitivity of CaMKII (*Figure 3E*). Supralinearity is indicated by the response to the combination of cAMP and calcium (Combo) being greater than the sum of responses to calcium and cAMP separately. ANCOVA

*Figure 8 continued on next page*

*Figure 8 continued*

results are in *Table 5*. (B) Summary of ppERK total activity in response to 300 s ITI. The contribution of the calcium pathway relative to the cAMP pathway is greatest when calcium is above threshold for an ultrasensitive CaMKII response. (C) Qualitative single pathway contribution to ppERK (ppERK from single pathway divided by ppERK from all pathways; sum of pathways exceeds 100%). The contribution of SynGap decreases with ITI, whereas other pathways increase with ITI.

The online version of this article includes the following figure supplement(s) for figure 8:

**Figure supplement 1.** Time course of inputs and key molecules upstream of ERK in response to four trains separated by 3, 20, and 300 s.

**Figure supplement 2.** Summary of ppERK total activity in response to each pathway.

**Figure supplement 3.** Time courses of ppERK and key molecules upstream of ERK in response to four trains of 100 Hz separated by 3 and 300 s.

**Figure supplement 4.** ERK activity versus ITI for several knockout experiments.

In summary, these results demonstrate that ERK signaling pathways integrate multiple inputs synergistically. In fact, ppERK activity in response to the combination of calcium and cAMP is always greater than the response to either calcium or cAMP alone. For high calcium inputs, as likely occur during L-LTP induction, these inputs combine supralinearly. Synergy is attributable to CaMKII phosphorylation of SynGap, as dispersion of SynGap from the spine enhances RasGTP and RapGTP in response to both calcium and cAMP inputs. Furthermore, our simulations suggest that Raf dimerization enhances Raf1 activation of ERK in response short ITI, thereby reducing the temporal sensitivity of ERK activation.

## Robustness

In computational models, it is critical to ensure the robustness of the results to variation in parameter values. The main result of the model is that ERK signaling pathways, no matter the protocol used, favor spaced stimuli. To assess the robustness of the temporal sensitivity of ERK signaling pathways to small variations in parameters, simulations were repeated using concentrations of ±10% of the control value. The concentrations were changed either individually or collectively with changes drawn from a uniform distribution (between −10% and +10%). These simulations show that total ERK increase with ITI, similar to control results. *Table 6* shows that the optimal ITI was 80 or 300 s (spaced) 80% of the time. We also evaluated the mean ppERK (across all ITIs) and the degree of temporal sensitivity, quantified as the difference between maximum ppERK and minimum ppERK (across ITIs). Results show that a 10% change in a single-molecule quantity rarely produced more than 20% change in mean ppERK (*Figure 10A*), with changes evenly distributed about 0%. In contrast, temporal sensitivity was more sensitive to parameter variations, with most parameter changes increasing temporal sensitivity (*Figure 10B*). An increase in temporal sensitivity with minimal change in mean ppERK implies that the response to short ITIs decreases while the response to long ITIs increases. As observed with single-molecule changes, small random changes to all molecule quantities increased temporal sensitivity (*Figure 10B*) with minimal changes in mean ppERK (*Figure 10A*).

To evaluate which molecule quantities produced the largest effect on ppERK, we used random forest regression to analyze the effect of random changes on the entire set of molecules. Random forest regression is a non-linear method (in contrast to linear regression) to determine which parameters are best for predicting the results. The algorithm creates numerous decision trees, each of which

**Table 5.** Two-way ANCOVA (analysis of covariance) results for ppERK AUC versus stimulation characteristics (ITI) and type (combination versus summation) as factors (N = 5 in each group).

| | cAMP+Gi$\beta\gamma$+calcium | | |
| --- | --- | --- | --- |
| | 0.5 $\mu$M ca | 18% PP1 | 50% PP1 |
| F (2,47) | 43.56 | 28.28 | 3.99 |
| P (ITI) | <0.0001 | <0.0001 | 0.023 |
| T (ITI) | 9.11 | 6.43 | 2.36 |
| P (type) | 0.05 | <0.0001 | 0.13 |
| T (type) | −2.04 | −3.90 | −1.56 |
| Conclusion | Significant for stim only | Significant for both type and stim | Significant for stim only |

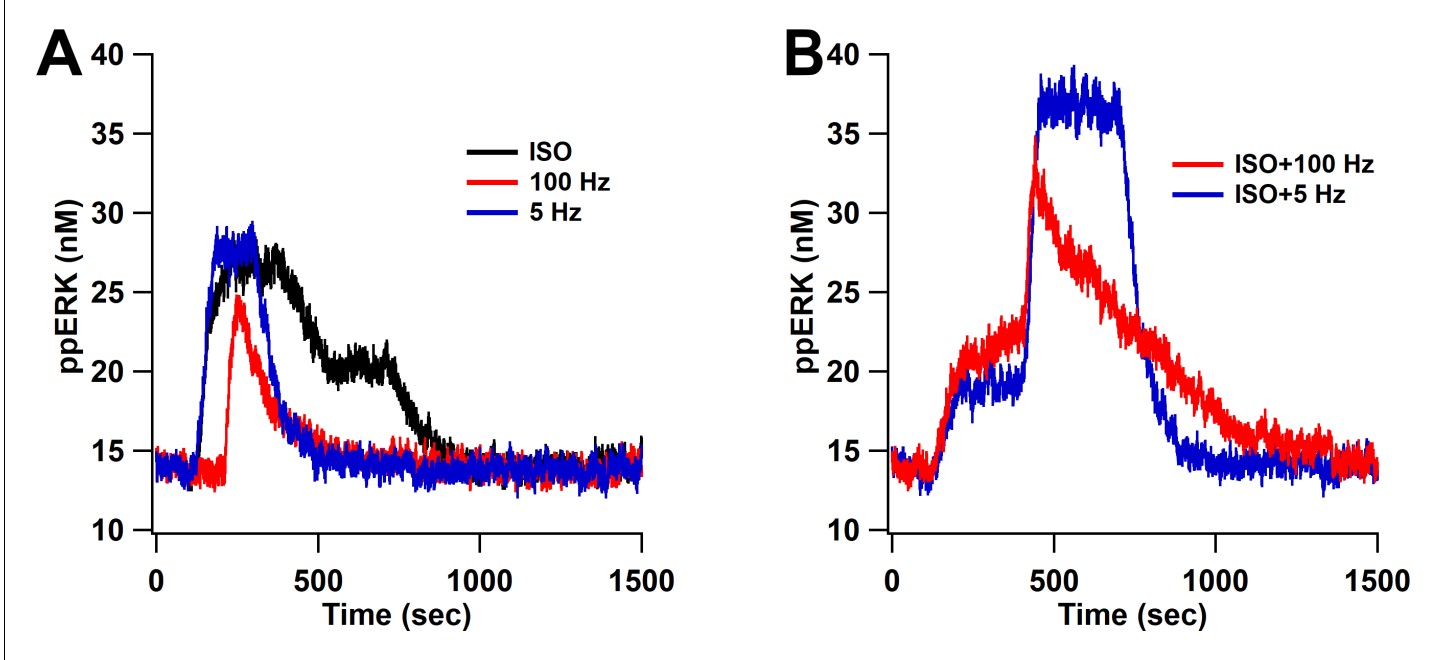

**Figure 9.** ppERK predicts the occurrence of L-LTP when bath-applied ISO is followed by either 1 s of 100 Hz stimulation or 180 s of 5 Hz. ppERK in response to isoproterenol (ISO), 1 s of 100 Hz, or 180 s of 5 Hz does not exceed 30 nM (**A**), whereas ppERK reaches or exceeds 35 nM in response to the combination of isoproterenol and either 100 Hz or 5 Hz stimulation (**B**).

The online version of this article includes the following figure supplement(s) for figure 9:

**Figure supplement 1.** Time course of inputs and key molecules upstream of ERK in response to ISO followed by 100 Hz or 5 Hz stimulation.

is a hierarchical set of rules, where each rule partitions the data (ppERK in our case) based on a single feature (molecule quantity in our case). The features of each tree are ranked based on their weight in accurately predicting the data. *Table 7* ranks the molecules, with the highest weight indicating the molecule that caused the most significant change in ppERK. The observation that most weights are small indicates that no single molecules can explain the change in ppERK. However, calcium pathway molecules such as RasGRF, calmodulin, pmca, and cAMP pathway such as Epac are ranked high, consistent with the contribution illustrated in single pathway simulations. In summary, the temporal sensitivity of ERK is quite robust to variation in parameters and favors spaced ITIs.

## Discussion

Numerous experiments have demonstrated that ERK is critical in L-LTP induction: ERK phosphorylates transcription factors, molecules involved in protein translation, and regulates cell excitability

---

**Table 6.** Count of best ITIs.

Count of preferred ITIs in robustness simulations, using either single-molecule changes of ±10% or random changes (between +10% and −10%) to all molecules. 80% of the parameter variations yield optimal ppERK in response to spaced stimulation (ITI = 80 or 300 s).

| ITIs | Random | 10% lower | 10% higher | Total percent |
|---|---|---|---|---|
| 3 | 38 | 0 | 0 | 4.69 |
| 20 | 46 | 0 | 1 | 5.80 |
| 40 | 81 | 3 | 1 | 10.49 |
| 80 | 154 | 112 | 51 | 39.14 |
| 300 | 181 | 40 | 102 | 39.88 |

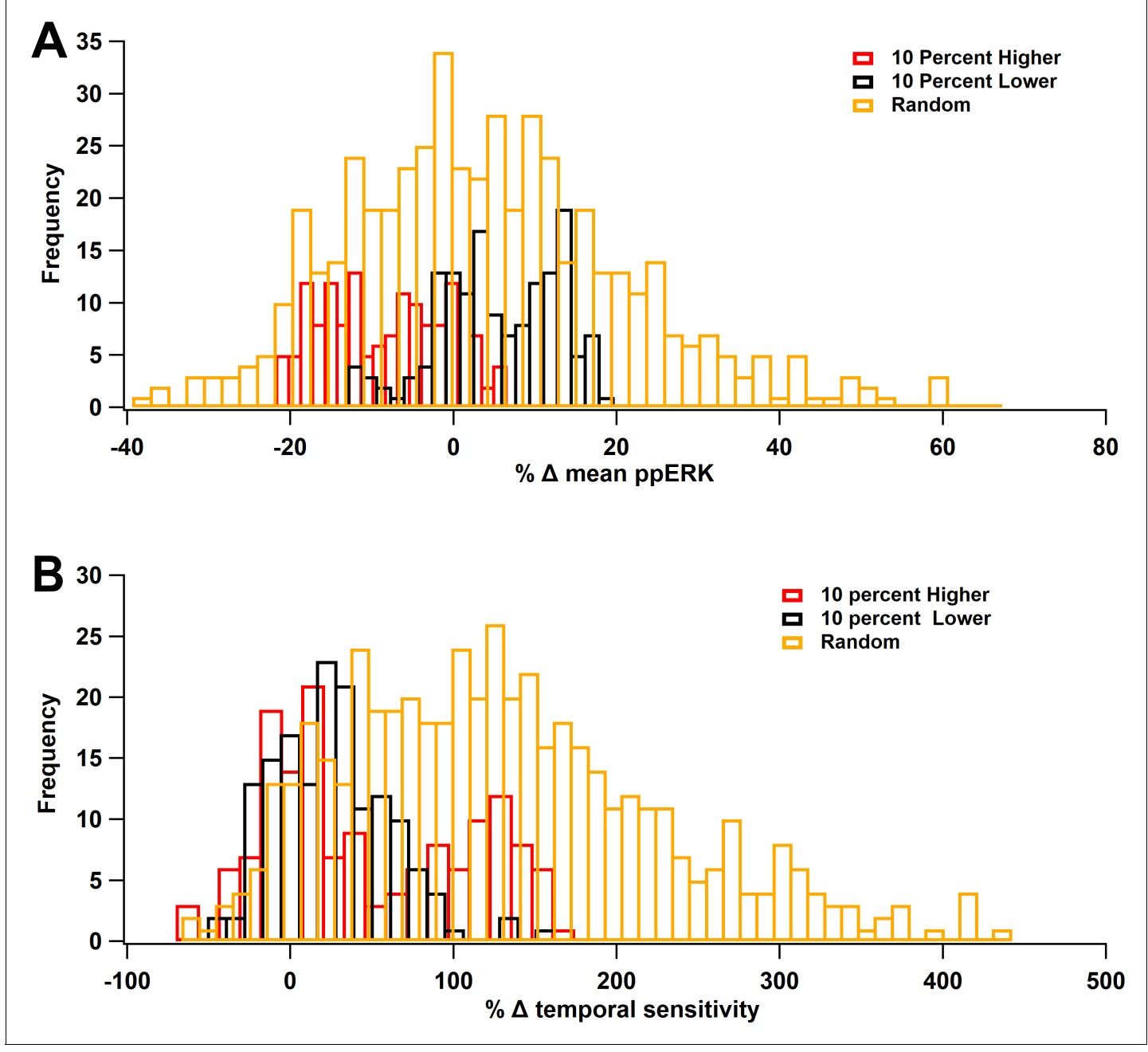

**Figure 10.** ERK Temporal sensitivity is robust to parameter variations. Molecule concentrations were changed individually by either +10% (red) or −10% (black) or the set of concentrations was changed randomly, within ±10% (gold). (A) Percent change in mean ppERK, averaged across all 5 ITIs. (B) Percent change in temporal sensitivity. Most variations increased temporal sensitivity.

through ion channel phosphorylation. ERK is activated through numerous pathways; however, it is unclear why so many pathways converge on ERK. Furthermore, some pathways are required, whereas other pathways can be compensated. Due to complicated interactions, models of signaling pathways are crucial for understanding the molecular foundations of L-LTP. Using a computational model of signaling pathways underlying L-LTP in hippocampus CA1, we evaluated temporal sensitivity and synergy between the ERK pathways utilized by different L-LTP stimulation protocols. Our data demonstrate that calcium and cAMP work synergistically to activate ERK, and we show for the first time the role of SynGAP in mediating the synergy between pathways. SynGAP mutations are associated with a variety of syndromes, such as autism and intellectual disability. Our simulations

**Table 7.** Random forest analysis.

One hundred random perturbations of all molecule concentrations (between +10% and −10%) were simulated. Features are ranked from most important to least important in predicting the temporal sensitivity. The sum of features' importance equals 1.

| Molecules abbreviation | Weights | Molecules full name |
|---|---|---|
| RasGRF | 0.084 | Ras guanine nucleotide–releasing factor |
| Sos | 0.074 | Son of sevenless |
| Cam | 0.068 | Calmodulin |
| MEK | 0.050 | Mitogen-activated protein kinase kinase |
| PDE2 | 0.050 | Phosphodiesterase 2 |
| pmca | 0.048 | Plasma membrane calcium pump |
| Epac | 0.046 | Exchange protein directly activated by cAMP |
| Calbin | 0.046 | Calbindin |
| CRKC3G | 0.044 | v-CRK Proto-Oncogene/Rap guanine nucleotide exchange factor 1 |
| rasGap | 0.043 | Ras GTPase-activating protein |
| SynGap | 0.042 | Synaptic rasGap |
| PP1 | 0.038 | Protein phosphatase 1 |
| Shc | 0.035 | Src homology collagen-like |
| Grb2 | 0.034 | Growth factor receptor-bound protein 2 |
| PP2A | 0.033 | Protein phosphatase 2 |
| MKP1 | 0.033 | Mitogen-activated protein kinase phosphatase 1 |
| Ng | 0.033 | Neurogranin |
| Src | 0.032 | Non-receptor tyrosine kinase |
| Cbl | 0.030 | Casitas B-lineage lymphoma |
| ERK | 0.027 | Mitogen-activated protein kinase |
| PDE4 | 0.025 | Phosphodiesterase 4 |
| PKA | 0.025 | Protein kinase A |
| NCX | 0.022 | Sodium calcium exchanger |

suggest that this phenotype might be in part caused by dynamics of ERK activation following closely the dynamics of SynGAP dispersion from the spine. Our data also show that stimuli spaced in time activate more ERK than stimuli massed in time, a temporal sensitivity similar to PKA but different than CaMKII. These ERK activation pathways are not redundant as each has a distinct dynamic, contributing to different time frames of ERK activation. For example, Epac contributes to an early transient ERK activation, whereas PKA contributes to a later transient ERK activation. With massed stimuli, calcium produces a rapid and sustained ERK activation, whereas with spaced stimuli, calcium only produces small transient ERK responses. The difference in dynamics and contribution of each pathway reinforce how neurons are able to recognize and discriminate different patterns in learning protocols.

The model elucidates that CaMKII contributes to ERK activation through phosphorylation of Syn-Gap. This synaptic RasGap, selectively expressed in the brain, regulates ERK activity by inactivating RasGTP and Rap1GTP (*Gamache et al., 2020*; *Pena et al., 2008*). CaMKII phosphorylation of Syn-Gap both increases its activity (*Walkup et al., 2015*) and causes dispersion to the dendrite (*Araki et al., 2015*). To investigate the role of dispersion, we repeated simulations without dispersion of pSynGap and observed that the ppERK amplitude was decreased by half. This effect is observed because its activity is doubled in the spine, reducing the available quantity of RasGTP and Rap1GTP. Thus, the dispersion of pSynGap attenuates its activity in the spine, which allows a long-lasting spine enlargement as observed in experiments (*Araki et al., 2015*). We also showed that blocking CaMKII binding to SynGap greatly reduced ppERK activity, whereas allowing CaMKII to bind but not phosphorylate SynGap greatly increased ppERK activity. This result is consistent with

experiments (*Rumbaugh et al., 2006*) showing that overexpression or depletion of SynGap significantly reduced and enhanced ERK activity, respectively. Thus, CaMKII-dependent SynGAP dispersion from the spine controls ERK recruitment.

One of our main conclusions is that feedforward loops can support ERK activation for L-LTP induction without PKC. Induction refers to the transient events that trigger the formation of L-LTP, while maintenance refers to the persistence of biochemical signaling that supports L-LTP. Our results showed that ppERK decays in less than an hour, suggesting that ERK is required during induction of L-LTP, but not during maintenance, which lasts for multiple hours. These results are consistent with experiments showing that inhibition of ERK 30 min after induction does not block LTP (*Kelleher et al., 2004*). PKMζ, the atypical form of PKC, participates in a feedforward loop to support L-LTP maintenance (*Ajay and Bhalla, 2004*; *Ajay and Bhalla, 2007*; *Tsokas et al., 2016*) however, the evidence that PKC is required for L-LTP induction is weak, and thus we did not include it in our model. Similarly, we did not include metabotropic glutamate receptors, which activate PKC, because structural plasticity, a correlate of LTP, is independent of mGluR stimulation (*Zhai et al., 2013*). An alternative feedforward loop involves PKA inhibiting ERK inactivation mechanisms (*Neves et al., 2008*), but PKA is not required for all forms of L-LTP induction. Our model reveals an additional feedforward loop, connecting CaMKII to SynGap to ERK, which is activated during L-LTP induction. This pathway may be critical for induction of L-LTP in CA1 for protocols resistant to inhibition of PKA and PKC (*Sweatt, 1999*). In summary, our study supports prior results showing that feedforward loops are sufficient to activate ERK but reveals a novel feedforward loop involving SynGap.

Temporal selectivity is an important characteristic of learning and memory formation. Animals can learn to respond similarly to a variety of temporal patterns or can respond differently depending on the temporal pattern (*Delamater and Holland, 2008*; *Mauk and Ruiz, 1992*). In some cases, synaptic input can produce either LTD or LTP depending on the temporal frequency of the input (*Chen et al., 2010*; *Hawes et al., 2013*). In other words, neurons can discriminate and respond differently to these different temporal input patterns (*Bhalla, 2017*), but also need to be able to learn despite a difference in temporal patterns. We showed that ERK is activated for a wide range of temporal input patterns, but what are the differences allowing temporal pattern discrimination?

One mechanism for temporal pattern discrimination is the temporal pattern of ERK itself, which can determine whether a cell differentiates or divides (*Santos et al., 2007*; *Sasagawa et al., 2005*; *von Kriegsheim et al., 2009*) and whether LTP and LTD occur (*Thiels et al., 2002*). We have summarized kinase activity as area under the curve, but some downstream molecules may be more sensitive to dynamics or peak activity. Indeed, our data show repeated transients of ppERK with 300 ITI versus a sustained ppERK with 80 and smaller ITIs, and the peak activity of ERK has a different temporal sensitivity than total activity. Total activity increases with ITI while peak decreases with ITI, implying that processes sensitive to peak versus AUC could distinguish these temporal patterns. We predict that these temporal patterns can be decoded by the nucleus to produce different gene expression patterns, as previously suggested (*Jain and Bhalla, 2014*). Our model prediction of different ERK dynamics could be tested using the new EKAR sensor to image ERK activation in hippocampal slices after L-LTP induction using different ITIs, and the prediction of different gene expression patterns could be tested using some of the newer gene expression techniques, e.g., TRAP-seq (*McKeever et al., 2017*), to determine whether short ITIs produced different gene expression patterns than long ITIs.

A second mechanism for temporal pattern discrimination is that pathways activating ERK have other targets essential for L-LTP induction. Our results show that the CaMKII and calcium contribution to ERK decreases with ITI, whereas the PKA and cAMP contribution to ERK increases with ITI. The opposite temporal sensitivity of these critical kinases, as previously reported (*Ajay and Bhalla, 2004*; *Ajay and Bhalla, 2007*; *Kim et al., 2010*; *Woo et al., 2003*), can contribute to pattern discrimination by the neuron, through activation of other downstream molecules that are required for LTP. For example, re-organization of the actin cytoskeleton (*Borovac et al., 2018*; *Honkura et al., 2008*; *Obashi et al., 2019*) is required to increase the size of the spine to allow for more glutamate receptors. In particular, both CaMKII and PKA control actin binding proteins, such as cofilin (*Havekes et al., 2016*; *Zhao et al., 2012*). Thus, even in cases where ERK activity cannot distinguish temporal patterns, the temporal sensitivity of other key kinases can contribute to temporal pattern sensitivity.

Our results suggest that numerous signaling pathways activate ERK to enhance sensitivity to a range of temporal patterns, but there are other differences between the pathways that suggest they are not redundant. If those pathways were redundant, then, contrary to experimental observations (*Gelinas et al., 2008b*; *Gelinas et al., 2008a*; *Scharf et al., 2002*; *Woo et al., 2003*), blocking one of them should not block ERK-dependent L-LTP. In addition, our model predicts that Rap1 and Ras are not interchangeable: Rap1 can compensate for Ras, but Ras cannot compensate for Rap1. Instead of redundancy, multiple pathways may be needed to control ERK dynamics and allow it to perform multiple tasks. For example, *Sasagawa et al., 2005* demonstrated that there are two distinct dynamics of ERK, transient and sustained, due to the distinct timeframe of Ras and Rap. Similarly, we observed that each ERK activation pathway has a distinct dynamic, each contributing to a different time frame of ERK activation. For example, Epac contributed to an early transient ERK activation, whereas PKA contributed to a later transient ERK activation, and CaMKII produced a sustained ERK activation. Thus, our model predicts that an Epac knockout would lead to slower ERK activation. These different time frames of ERK may phosphorylate different targets (*Keyes et al., 2020*; *Zhai et al., 2013*; *Zhang et al., 2018*). For example, Epac, a molecule of emerging importance in neurodevelopmental disorders, may have a role in disorders in cAMP degradation by enhancing the early activation of ERK. SynGap may be critical for a pool of ERK in the spine, whereas PKA may be essential for a pool of ERK that translocates to the nucleus. In this context, we predict that overexpression of SynGap would not reduce ERK activity in the nucleus but only in the spine. An experimental test of this prediction could use EKAR imaging (*Harvey et al., 2008*) to quantify ERK expression in the spine versus nucleus or measure gene expression with overexpression of SynGap.

Different temporal patterns may produce different outcomes by activating different pools of ERK. Studies have shown that scaffolding proteins, such as Kinase Suppressor of Ras (KSR) protein (*Dougherty et al., 2009*; *Shalin et al., 2006*) and β-arrestins (*Bourquard et al., 2015*; *DeFea et al., 2000*), can create multi-protein complexes of ERK, MEK, and Raf and dictate the subcellular location of ERK activity. Thus, a critical question is whether the spatial pool of ERK depends on which signaling pathways are activated during induction of LTP. These different pools of ERK may perform different functions, such as translocating to the nucleus to initiate gene transcription, staying in the cytoplasm to initiate local changes in excitability, or allowing stimulated spines to capture newly synthesized proteins to enable persistence of L-LTP (as in synaptic tagging and capture) (*Frey and Morris, 1997*; *Redondo et al., 2010*; *Sajikumar et al., 2007*; *Young et al., 2006*). This leads to a second question regarding what limits the spatial spread of molecules involved in marking spines as having been stimulated. In this context, a possible role for SynGAP dispersion would be to limit diffusion of active Ras family proteins to nearby, non-stimulated spines, thereby creating microdomains of ERK. Spatial models (*Bhalla, 2017*; *Kim et al., 2011*) have shown that buffering and other inactivation mechanisms can produce spine-sized microdomains of calcium and cAMP, but not PKA or CaMKII, suggesting additional mechanisms may be needed to produce spine-sized microdomains of ERK. Taking those questions together it will be quite interesting to implement a spatial model to investigate mechanisms that govern spatial pools of ERK and spatial and tagging specificity.

## Materials and methods

### Computational methods

To investigate how temporal pattern of synaptic activity determines which pathway in the ERK signaling cascade (*Figure 1A*, *Figure 1—source data 1–5*) dominates in dendritic spines of hippocampal CA1 pyramidal neurons, we developed a single-compartment, stochastic reaction–diffusion model of pathways activating ERK. The model was built by merging and adapting two existing models of synaptic plasticity in hippocampal neurons. One of those models (*Jędrzejewska-Szmek et al., 2017*) explains how different temporal patterns activate different kinases such as CaMKII and PKA that are critical for L-LTP induction, and the other model (*Jain and Bhalla, 2014*) demonstrates that protein synthesis is the result of synergistic activation of different pathways such as ERK and mTOR to generate unique patterns in response to different L-LTP stimuli. These previously published models were modified by adding several molecules and reactions critical for ERK activation.

One major change to the merged model is C-Raf dimerization. Dimerization regulates C-Raf activation and its subcellular localization (*Desideri et al., 2015*; *Garnett et al., 2005*). Full activation of C-Raf, but not BRaf, requires dimerization. RasGTP binds to C-Raf to form an inactive complex, C-Raf-RasGTP, which then homodimerizes to become active, thus able to phosphorylate MEK that activates ERK. Another major change is the addition of two important molecules, RasGRF and Syn-Gap, involved in hippocampal LTP (*Araki et al., 2015*; *Darcy et al., 2014*; *Jin and Feig, 2010*; *Kim et al., 1998*; *Li et al., 2006*). RasGRF is activated by binding to calcium-calmodulin; once activated, RasGRF binds to RasGDP and allows the exchange of GDP for GTP. SynGap inactivates Rap1GTP and RasGTP with low affinity. Once phosphorylated by CaMKII, SynGAP affinity doubles toward RasGTP and Rap1GTP and causes SynGAP dispersion from synapses to the dendrite.

Initial conditions and rate constants are either taken from previous models (*Jain and Bhalla, 2014*; *Jędrzejewska-Szmek et al., 2017*) or adjusted to reproduce experimentally measured concentrations of molecules (*Figure 1—source data 1–6*) or to match time course data, for example rate constants governing CaMKII autophosphorylation. The concentration of some species was adjusted to produce an intracellular concentration of calcium of ~50 nM and cAMP of ~30 nM, among others. For example, the concentration of PDEs was adjusted to produce a 30 nM basal cAMP. Calcium extrusion mechanisms and the low affinity and immobile calcium buffer were adjusted to obtain a steady state of about 50 nM of internal calcium. To match the in vitro data on basal concentration, we ran the model for about an hour to obtain steady-state concentrations for all molecules.

Several simulations use a compartment with a volume comparable to a spine head (about 0.1–0.8 $\mu m^3$). To minimize noise inherent in stochastic models, most simulations used a larger compartment, of volume 4 $\mu m^3$. Other than the level of noise, no differences in temporal sensitivity were observed between the small and large volume models.

## Stimulation protocols

In brain slices, presynaptic stimulation results in calcium influx through NMDA receptors, and norepinephrine (NE) or dopamine, which bind to Gs-coupled metabotropic receptors (*Jędrzejewska-Szmek et al., 2017*) is required for L-LTP (*Huang and Kandel, 1995*). To investigate the effect of calcium distinct from cAMP produced by NE, the model is activated using three independent inputs leading to ERK activation: calcium (through RasGRF and SynGap), cAMP (through Epac and PKA), and βγ subunit of inhibitory G protein (Gi) (through implicit PKA phosphorylation of βAR). βARs are mostly coupled with stimulatory G protein (Gs). However, when phosphorylated by PKA, βARs decouple from Gs and couple with inhibitory G proteins (Gi). Both Gs-activated and βAR coupled with Gi-activated signaling pathways converge to activate ERK using different pathways. To determine whether maximal ERK activation requires multiple pathways, the contribution of each pathway was tested singly and in combination using either a duration or amplitude protocol (*Table 1*). Duration protocols used a sustained concentration of calcium, cAMP, and Giβγ. Amplitude protocols used transient (1 s duration) elevations of different concentrations of the inputs. Calcium and cAMP each have several different pathways to ERK activation; thus, to determine the contribution of each of the (sub)pathways, the above simulations were repeated with one of the rates set to zero, as shown in *Table 3*. To investigate whether different L-LTP temporal patterns select different signaling pathways for ERK activation, simulations were performed using stimulation protocols identical to the experimental protocols used to induce L-LTP. Therefore, we stimulated the model with four trains of 100 Hz for 1 s at five different intertrial intervals: 3, 20, 40 s (massed) or 80 and 300 s (spaced) (*Figure 1C*). Each stimulation pulse triggered a transient elevation of either calcium, cAMP (direct action and in combination with Giβγ) singly, or in combination.

We also tested different forms of stimulation patterns that induce L-LTP. The additional stimulation protocols are bath-applied ISO followed by 1 train of 100 Hz (ISO+100 Hz) or 5 Hz (ISO+5 Hz). The inputs used to run these simulations are derived from simulations of a spatial model response to glutamate (*Jędrzejewska-Szmek et al., 2017*).

## Simulation and analysis

The model is implemented using a stochastic reaction–diffusion simulator NeuroRD (*Jędrzejewski-Szmek and Blackwell, 2016*), version 3.2.4 (*Jędrzejewski-Szmek and Blackwell, 2018*), using the

adaptive (asynchronous tau-leap) numerical method and tolerance of 0.1. Even though stochastic fluctuations observed using small compartments do not impact the results, the stochastic algorithm is extremely fast, especially for stiff systems; thus, there would be no advantage to switching to a potentially less accurate deterministic simulator. All model files needed to run the simulations, including molecule quantities, reaction rate constants, and stimulation quantities, are available on GitHub (*Miningou Zobon et al., 2021*) and on modelDB, accession number 267073.

Biochemical experiments assess kinase activity by measuring the change in concentration of the product after a specific time. Thus, in these simulations, the kinase activity is quantified as total activity over simulated time compared to the activity under conditions of no stimulation, also known as area under the curve (AUC). To account for stochastic variation, simulations were repeated five times (each using a different random seed) and results presented are the mean and standard error over the trials. The sample size is based on previous publications (*Jędrzejewski-Szmek and Blackwell, 2016*), which found this sample size sufficient to show differences between conditions. Analysis was done in Python 3, using nrdh5_analV2.py available on GitHub (*Blackwell and Miningou Zobon, 2021*).

Statistical analysis was done in Python3 (https://github.com/neurord/ERK/tree/master/Analysis), *Miningou Zobon and Blackwell, 2021*. To assess whether the response was linear, the AUC versus stimulation duration or concentration was fit to a line, a Hill equation, and a logarithmic equation (the latter two selected subjectively as matching the shape of the curve). The response was considered non-linear if both adjusted $R^2$ and Akaike information criteria were better for one of the non-linear equations. To determine whether two pathways combined linearly, analysis of variance was used, with AUC of doubly ppERK as the dependent variable, and intertrial interval and stimulation type (combination or summation) as independent variables (N = 5 in each group). Random forest regression was used to analyze the robustness simulations, to determine which parameter values had the most influence on the change in ppERK AUC. Random forest is a non-linear method of regression that creates multiple decision trees. Each tree is made by randomly selecting features, for example molecule concentration, that best predict the dependent variable, that is ppERK. By using a large number of decision trees (100 for this analysis), the results are quite robust. The outcome of the random forest is a weight for each feature, indicating its usefulness in predicting the dependent variable. By design, the sum of weights for all features equals 1.

## Acknowledgements

This work was supported through the joint NIH-NSF CRCNS program through NSF grant 1515686 and NIMH grant R01MH 117964.

## Additional information

### Funding

| Funder | Grant reference number | Author |
| --- | --- | --- |
| National Institutes of Health | R01MH 117964 | Kim T Blackwell |
| National Science Foundation | 1515686 | Kim T Blackwell |

The funders had no role in study design, data collection and interpretation, or the decision to submit the work for publication.

### Author contributions

Nadiatou T Miningou Zobon, Conceptualization, Formal analysis, Investigation, Visualization, Methodology, Writing - original draft, Writing - review and editing; Joanna Jędrzejewska-Szmek, Writing - review and editing, model simulations; Kim T Blackwell, Conceptualization, Software, Formal analysis, Supervision, Funding acquisition, Investigation, Methodology, Writing - original draft, Project administration, Writing - review and editing

## Author ORCIDs

Nadiatou T Miningou Zobon (iD) https://orcid.org/0000-0002-5824-7706
Joanna Jędrzejewska-Szmek (iD) https://orcid.org/0000-0002-2336-0848
Kim T Blackwell (iD) https://orcid.org/0000-0003-4711-2344

## Decision letter and Author response

Decision letter https://doi.org/10.7554/eLife.64644.sa1
Author response https://doi.org/10.7554/eLife.64644.sa2

# Additional files

## Supplementary files

- Transparent reporting form

## Data availability

All model files are freely available on https://github.com/neurord/ERK/releases/tag/1.0.0 (copy archived at https://archive.softwareheritage.org/swh:1:rev:27db0c5f79d5a1a7922b06e2a0b258102-dec9670). All programs to analyze simulation output are available on https://github.com/neurord/NeuroRDanal/releases/tag/2.0.0 (copy archived at https://archive.softwareheritage.org/swh:1:rev:4fe66de8f9be91a32109fcd5db862a8bac9371e3). Programs for the statistical analysis and random forest analysis are available on https://github.com/neurord/ERK/tree/master/Analysis (copy archived at https://archive.softwareheritage.org/swh:1:rev:27db0c5f79d5a1a7922b06e2a0b258102dec9670). These URLs are provided in the manuscript methods section. Model files are available from modelDB, accession number 267073.

The following dataset was generated:

| Author(s) | Year | Dataset title | Dataset URL | Database and Identifier |
|---|---|---|---|---|
| Miningou ZNT, Blackwell KT | 2021 | Model files for simulating the ERK signaling pathway during L_LTP induction in the hippocampus | https://senselab.med.yale.edu/modeldb/enterCode?model=267073 | modelDB, 267073 |

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
