## [Decision Letter]

**Acceptance summary:**

It is interesting how a few molecules, such as the protein kinase ERK, act as a convergence points for multiple pathways in synaptic plasticity. In this study the authors show how ERK not only acts like a hub for many inputs, but also selectively responds to different time-patterns of inputs depending on which combination of inputs are active.

**Decision letter after peer review:**

Thank you for submitting your article "Temporal pattern and synergy influence activity of ERK signaling pathways during L-LTP induction" for consideration by *eLife*. Your article has been reviewed by 3 peer reviewers, one of whom is a member of our Board of Reviewing Editors, and the evaluation has been overseen by Ronald Calabrese as the Senior Editor. The reviewers have opted to remain anonymous.

The reviewers have discussed the reviews with one another and the Reviewing Editor has drafted this decision to help you prepare a revised submission.

As the editors have judged that your manuscript is of interest, but as described below that additional simulations and analyses are required before it is published, we would like to draw your attention to changes in our revision policy that we have made in response to COVID-19 (https://elifesciences.org/articles/57162). First, because many researchers have temporarily lost access to the labs, we will give authors as much time as they need to submit revised manuscripts. We are also offering, if you choose, to post the manuscript to bioRxiv (if it is not already there) along with this decision letter and a formal designation that the manuscript is "in revision at eLife". Please let us know if you would like to pursue this option. (If your work is more suitable for medRxiv, you will need to post the preprint yourself, as the mechanisms for us to do so are still in development.)

Summary:

This study takes on the question of the roles of the many pathways leading to ERK activation in long-term potentiation. This is an advance since few models consider timing roles of more than a couple of input pathways to ERK or in plasticity. The authors consider two aspects: how pathways sum to give strong responses, and distinct temporal pattern selectivity. They show that both summation linearity, and pattern selectivity, are strongly governed by which pathways are engaged in driving the response.

Essential Revisions:

A key point discussed by the reviewers was whether the temporal pattern selectivity represented sufficient interest and novelty. Could the authors make a strong case for this? In addition the reviewers felt that the following points would be essential to address in a revision:

1. Could the authors provide extensive comparisons to experiment? All the reviewers felt that the parameterization was inadequately validated, specially as the model was a significant change from its original sources.

2. Could the authors bring a more complete account of activation of the chosen pathways by synaptic input? It should be possible to simulate how all the pathways are triggered, presumably in an overlapping way, by different kinds of input.

3. While ERK is important, there are numerous other pathways that play a key role in plasticity. The authors should specify the role of ERK and relate it to other pathways.

4. The analyses of summation and linearity should be done in a more statistically complete manner.

5. The authors should provide better documentation of their reaction system through diagrams and standard formats.

6. What testable predictions does the model make?

In addition, the reviewers brought up other relevant points in their individual reviews.*Reviewer #1:*

This study takes on the question of the roles of the many pathways leading to ERK activation in long-term potentiation. This is an advance: few models consider more than a couple of input pathways. The authors consider two aspects: how pathways sum to give strong responses, and distinct temporal pattern selectivity.

The model and analysis is potentially interesting, but the paper would be much strengthened if there were more convincing validation of the properties of the model by way of simulations to compare with experiments. I also bring up a couple of points about how better to link synaptically-driven responses to the properties of the model.

1. I was looking to see some model validation before diving into the predictions of the model. Specifically, it would have been useful to compare the model ERK responses to each of the individual stimulation pathways that the authors implement.

Around line 531 the authors describe how parameters were chosen. These seem to rely on previous work or on steady-state levels in the absence of stimulation. Since the model is quite different from the source models, it is important to provide additional validation. Further, time-series responses and dose-response curves give far tighter constraints on system behavior than single-point steady-state readouts. A figure with a range of such validation runs would be very helpful.

1b. There is a nice analysis (Table 2) of effects of KO on responses. Surely there are physiological experiments to compare with this? In the para around line 137 several experiments are mentioned but it would be useful to show simulations to compare with the data.

2. What is the relative strength of contributions by each of the proposed 5 pathways to ERK regulation? How was this ascertained?

3. It is interesting to see differences in linearity between cAMP and Ca pathways. However, these are themselves downstream of the synaptic input. Could the authors explore how these differences would manifest upon synaptic input to a typical synapse having AMPAR, NMDAR, and mGluR? Won't they be rather overlaid on each other?

4. While it is nice to see a model with as many as 5 pathways driving ERKII, I am curious about the choice of these input pathways. From synaptic input one would expect to see an mGluR component. Do the authors envision that this folds into Gbg? Further, one sees other signals such as BDNF as major players in plasticity.

4b. I bring this up also in the context of the reported time-courses. Experimental time-courses of other pathway inputs can be quite different from those explored here, BDNF tending to be quite slow.

Reviewer #2:

There are, however, a couple of points that I feel should be addressed, in order for me to enthusiastically recommend this manuscript for publication.

1. There needs to be additional technical detail on how the original models were expanded. The model presented here was developed by merging Jȩdrzejewska-Szmek et al., 2017 and Jain and Bhalla, 2014 models. These models were developed based on experimental data and validated with independent experimental datasets in a rigorous manner. It is not clear how the combining these two models, and the additional molecules and reactions added have affected the dynamics of ERK activation, and how comparable they are to the original experimental data used for model development in the previous modeling efforts. It is not clear if the model was reparametrized.

2. Beyond the ERK activation traces, it would be useful for clarity sake to also include the simulated traces for the activation of the upstream molecules (PKA, RAS, RAP, etc). Given how additional changes have been made additional information should be provided to ensure that the contribution of each pathway is accurately represented.

*Reviewer #3:*

This paper is primarily about modeling the ERK pathway during the induction of synaptic plasticity. This pathway has been previously modeled, and this is cited in the paper. The main addition here is the addition of the effect of SynGap which is necessary in some form of LTP. This is a very detailed study, and what it seems to primarily show is that the ERK pathway favored spaced vs. massed stimulation protocols. This is a very detailed paper, but no conceptually new ideas are presented here. The paper adds to an existing foundation, but fails to make the case that this is a very significant addition. What is the significant consequence at a higher level of these added details?

The ERK pathway is just one component of a much larger set of pathways that control synaptic plasticity, how much do we learn from studying this pathway in isolation? Also, the paper cites the importance of this pathway to L-LTP, is it the induction phase of L-LTP? It seems so because ppRRK decays in less than an hour. How then does this pathway contribute to the maintenance of L-LTP, these processes such as a possible upregulation of protein synthesis are not part of this model either.

This paper studies in detail different pathways that influence ERK activation is synapses. This is a very detailed study, but how many details do we actually know? For a detailed paper though it seems that many of the details are missing. Is there a detailed diagram of reactions, or set of equations for all these reactions? Some coefficients are named in figure 1, and this might be sufficient for a schematic description of the model in the paper, however there must be somewhere a detailed description of all reactions. How many species are there here, how many coefficients? How are coefficient values known? How many coefficients are directly estimated? The paper does carry out an extensive robustness analysis, though it is not well explained.

What are the major takeaways from this paper, and what experiments could test this model?

To summarize, the paper is very detailed carefully constructed and executed, it fails to convince that the problem it addresses is very significant, and it makes no conceptual breakthroughs.

The induction protocols seem to include not only calcium pulses, but also cAMP and g-protein coupled pulses, what is the cellular origin of these other pulses? Is the G protein activation due to β-adrenergic activation, is this activation necessary for plasticity? What is the cellular origin of the cAMP and Gi activation in slice? Similarly, what primarily activated cAMP in real synapses?

Is ERK required for all forms of LTP? Some references indicate it might not be. Is ERK necessary for early LTP induced by 100Hz stimulation?

I get lost in the details of the introduction, description of different pathways should be in results and relate to figure 1. It looks like a shopping list, why is this significant?

Why is it important that this is a stochastic diffusion reaction model rather than a deterministic model? Nothing about fluctuations, except for small error bars are mentioned in this paper.

Do not reference Zhang et al. Nature Neuro 2012, its' not a mammalian LTP model, still there are similarities (and differences), especially as ERK pathways are studied and since it relates to spaced learning.

229 – where do we see autophosphorylation in figure 3F?

Figure 6B – hard to see shape of symbols.

No model diagrams or equations here, for example how in CaMKII autophosphorylation implemented? Figure 6C – what is old CaMKII? They give a reference, but we do not know what the new model is.

Narrow picture by looking only at ERK. – For example, increasing PP1 can enhance ppERK, however it might reduce overall LTP due to dephosphorylation of AMPA receptors. What wins?

Figure 7 – what in figure 7 shows super linearity? It's in the title, where else? This applies to paragraph starting in line 368 as well. Where do we see these results. We can try to eyeball it. It would be useful to show a plot of combined model/(sum of individual models). By eyeballing this it seems (but might be wrong) that the super-linearity is minimal.

403 – the first time random-forest was mentioned in line 403, and this method is not explained or defined prior to this or even in this paragraph. For *eLife* type of readers this needs to be explained in simple terms.

423 – typo – "Molecules were randomly simultaneous(ly)"

What to weights in table 5 mean? This relates to the random forest analysis which is not explained.

Similar problems with figure 8, what is shown here? What is the analysis used here? This is totally unclear. Are molecules here changed one at a time? How much are they changed? Is this the 10% discussed earlier? It would also be more instructive to show relative range changes.

How is this paper related to the recent Maki-Marttunen paper from the same lab which appeared recently (https://www.biorxiv.org/content/10.1101/2020.01.27.921254v1.abstract), it is not referenced here. That is a more general paper but seems to exclude ERK. If ERK is so significant how can it be excluded in one paper and included in another from the same lab?

Discussion starts OK, but again devolves into excessive details.

Does this model possibly explain induction of L-LTP, or its maintenance phase? It seems the prior since ppERK activity returns to baseline; these should be distinguished.

In summary – this paper fails to convince that it is of general interest beyond a narrow community. However, the comments in this section show that there are changes to be made even for the specific community of scientists interested in the signaling pathways of LTP.

[Editors' note: further revisions were suggested prior to acceptance, as described below.]

Thank you for resubmitting your work entitled "Temporal pattern and synergy influence activity of ERK signaling pathways during L-LTP induction" for further consideration by *eLife*. Your revised article has been evaluated by Ronald Calabrese (Senior Editor) and a Reviewing Editor.

The manuscript has been improved but there are some remaining issues that need to be addressed, as outlined below:

Summary points from the previous decision:

Point 1: Comparisons to experiments and validation of parameterization.

The authors have partly addressed this point, with comparisons of simulations to several additional experiments. They have also provided tables of data sources but many parameters are linked to previous models rather than to experiments. It would be helpful if the authors could explicitly address the point about _parameter_ validation, and to what extent systems level experiments provide validation for rate terms.

Point 2: Activation of pathways by synaptic input.

Here the authors use the model from their 2017 paper to generate Ca and cAMP waveforms, and report results consistent with experiments. It is just a single figure though, and only reports ppERK. We don't see the upstream pathways and their responses. Since the authors have now added these upstream pathways, it would be good to see the time-courses for upstream key molecules indicated in figure 1. These include Ca, Gbg, cAMP, as well as CaMKII, PKA, synGap, Ras and Rap1. It would be useful to have a figure with these time-courses for the major stimulus patterns used in the figures. In Figure 8 – supplementary Figure 2B the authors provide some time-courses for a few molecules. A complete set would be desirable.

Point 4: Linearity. Here the authors report ANCOVA analysis for figures 3, 5, 6. This addresses the point.

Point 5: Source data now provided. Unfortunately it isn't in SBML. If the authors can apply a conversion program to their model, an SBML version would be valuable.

Point 6: Predictions: These are now provided.

Responses to reviewer 1: OK.

Responses to reviewer 2: OK

Responses to reviewer 3: We recommend that the response to the point about induction protocols would be best addressed with graphs of the time-courses mentioned above in Point 2.

Also, the response to the reviewer's point about the use of a stochastic method would be good to include in the text. We did not see this statement there.

---

## [Author Response]

Summary:This study takes on the question of the roles of the many pathways leading to ERK activation in long-term potentiation. This is an advance since few models consider timing roles of more than a couple of input pathways to ERK or in plasticity. The authors consider two aspects: how pathways sum to give strong responses, and distinct temporal pattern selectivity. They show that both summation linearity, and pattern selectivity, are strongly governed by which pathways are engaged in driving the response.Essential Revisions:A key point discussed by the reviewers was whether the temporal pattern selectivity represented sufficient interest and novelty. Could the authors make a strong case for this?

We now begin the manuscript with a general explanation of the importance of temporal pattern (lines 36-43).

“Temporal patterns are a key feature of the environment. The speed of traversing space is indicated by the time between spatial cues. In classical conditioning, adaptive changes in behavior require an animal to respond to a cue in an appropriate time frame to gain a reward or avoid punishment (Delamater and Holland, 2008; Mauk and Ruiz, 1992). In these and other tasks, environmental cues, i.e., sensory inputs, are converted into spatio-temporal patterns of activation. Pattern discrimination requires that neurons are able to discriminate and respond differently to these different temporal input patterns (Bhalla, 2017); conversely, neurons need to be able to learn despite a difference in temporal patterns.”

We also explain the relevance of temporal pattern discrimination for LTP (lines 46-51) and other cell functions (lines 83-88). In summary, temporal pattern of synaptic inputs can influence the response of the neuron by controlling the temporal pattern of ERK activation. Temporal pattern has been shown to affect numerous aspects of neuron function, including whether LTD or LTP occurs. In other word, temporal pattern and timing of the inputs regulate learning and memory.

We discuss the importance of temporal pattern in the discussion, lines 444-492.

1. Could the authors provide extensive comparisons to experiment? All the reviewers felt that the parameterization was inadequately validated, specially as the model was a significant change from its original sources.

We did extensive comparisons to experiments in the first version, but did not report them. We have now added several of those comparisons as well as several new validations.

We have added graphs of RasGTP in response to inputs of 60 sec train of 0.5 μm of calcium and directly compare to Harvey et al., 2008 (lines 108-112, Figure 2A).

We have added graphs of ppERK in response to inputs of 1 sec train of 5 μm and 1 μm of calcium and cAMP respectively and directly compare to Kasahara et al., 2001 (lines 112-116, Figure 2B).

We simulated the L-LTP protocols induced with isoproterenol followed by either 100 Hz or 5 Hz stimulation, and show that both of those protocols produce a higher ppERK than ISO, 100 Hz or 5 Hz alone (lines 309-319, Figure 9).

We simulated knockout experiments of bRaf, Raf1, Ras and Rap (lines 320-339 and Figure 8 – Supplementary Figure 3), and show that our results are consistent with experimental data.

2. Could the authors bring a more complete account of activation of the chosen pathways by synaptic input? It should be possible to simulate how all the pathways are triggered, presumably in an overlapping way, by different kinds of input.

Indeed, these pathways are triggered by both synaptic and neuromodulator input. Neuromodulators such as dopamine and norepinephrine are released during LTP stimulation and activate Gs coupled receptors to produce cAMP. Calcium influx through NMDA receptors not only activates the calcium pathways in our model but also can activate adenylyl cyclases to produce cAMP. To better investigate ERK activation during L-LTP induction, we have generated calcium and cAMP input by simulation of our published model of those pathways (Jȩdrzejewska-Szmek et al., 2017), and then measured the ERK response to calcium and cAMP using several LTP induction protocols. These simulations were done by Joanna Jedrzejewska-Szmek, who has been added as a co-author. These results, lines 309-319 and new Figure 9, show that isoproterenol followed by either 100 Hz or 5 Hz produce greater ppERK than ISO, 100 Hz or 5 Hz alone, consistent with the experimental observation that the combination stimuli are needed to produce L-LTP.

3. While ERK is important, there are numerous other pathways that play a key role in plasticity. The authors should specify the role of ERK and relate it to other pathways.

We have clarified the role of ERK in the introduction (lines 61-62) and we discuss its role further in the discussion, lines 380-382. We have related ERK to other pathways in the context of temporal pattern sensitivity, lines 451-483. Specifically, both PKA and CaMKII phosphorylate molecules involved in cytoskeletal reorganization (lines 465-469).

4. The analyses of summation and linearity should be done in a more statistically complete manner.

We now added statistical analysis of linearity for all graphs of ppERK versus duration and concentration. We do a statistical analysis of whether the combination of molecules produces linear summation for all combinations. The statistics have been added to the captions of figures 3, 5, 6, or as additional tables.

5. The authors should provide better documentation of their reaction system through diagrams and standard formats.

We now provide a table of reactions, reaction rates, and molecule quantities, Figure 1 Source Data 1-6.

6. What testable predictions does the model make?

We have now added several predictions to the discussion.

We predict that these temporal patterns can be decoded by the nucleus to produce different gene expression patterns. Our model prediction of different ERK dynamics could be tested using the new EKAR sensor to image ERK activation in hippocampal slices after L-LTP induction using different ITIs, and the prediction of different gene expression patterns could be tested using some of the newer gene expression techniques, e.g.TRAP-seq (McKeever et al., 2017), to determine whether shorter ITIs produced different gene expression patterns than long ITIs (lines 452-458).

We observed that each ERK activation pathway has a distinct dynamic, each contributing to a different time frame of ERK activation. For example, Epac contributed to an early transient ERK activation, whereas PKA contributed to a later transient ERK activation, and CaMKII produced a sustained ERK activation. Thus, our model predicts that an Epac knockout would lead to a slower ERK activation (lines 480-484).

These different time frames of ERK may phosphorylate different targets (Keyes et al., 2020; Zhai et al., 2013; Zhang et al., 2018). For example, Epac, a molecule of emerging importance in neurodevelopmental disorders, may have a role in disorders in cAMP degradation by enhancing the early activation of ERK. SynGap may be critical for a pool of ERK in the spine, whereas PKA may be essential for a pool of ERK that translocate to the nucleus. In this context, we predict that overexpression of SynGap would not reduce ERK activity in the nucleus but only in the spine. An experimental test of this prediction could use EKAR imaging (Harvey et al., 2008a) to quantify ERK expression in the spine versus nucleus or measure gene expression with overexpression of SynGap (lines 484-492).

Reviewer #1:1. I was looking to see some model validation before diving into the predictions of the model. Specifically, it would have been useful to compare the model ERK responses to each of the individual stimulation pathways that the authors implement.

As explained above, in essential revisions, we have performed additional simulations to validate the model:

We did extensive comparisons to experiments in the first version, but did not report them. We have now added several of those comparisons as well as several new validations.

We have added graphs of RasGTP in response to inputs of 60 sec train of 0.5 μm of calcium and directly compare to Harvey et al., 2008 (lines 108-112, Figure 2A).

We have added graphs of ppERK in response to inputs of 1 sec train of 5 μm and 1 μm of calcium and cAMP respectively and directly compare to Kasahara et al., 2001 (lines 112-116, Figure 2B).

We simulated the L-LTP protocols induced with isoproterenol followed by either 100 Hz or 5 Hz stimulation, and show that both of those protocols produce a higher ppERK than ISO, 100 Hz or 5 Hz alone (lines 309-319, Figure 9).

We simulated knockout experiments of bRaf, Raf1, Ras and Rap (lines 320-339 and Figure 8 – Supplementary Figure 3), and show that our results are consistent with experimental data.

Around line 531 the authors describe how parameters were chosen. These seem to rely on previous work or on steady-state levels in the absence of stimulation. Since the model is quite different from the source models, it is important to provide additional validation. Further, time-series responses and dose-response curves give far tighter constraints on system behavior than single-point steady-state readouts. A figure with a range of such validation runs would be very helpful.

We have clarified to explain that we do not rely on single steady-state levels for selecting parameters. We do constrain our responses according to time-series data (e.g. for pCaMKII, and for Epac) and dose-response curves (e.g. for CaMKII activation). But we also use basal concentrations as constraint, which is not always done. We now add tables of parameters which shows that most parameters were taken from the two models and identifies the parameters that were newly derived or re-parameterized. We provide the URL to the code on github that implemented parameter optimization for CaMKII and SynGap. In addition, we provide additional validation figures comparing model responses with published data (for ppERK and RasGTP).

1b. There is a nice analysis (Table 2) of effects of KO on responses. Surely there are physiological experiments to compare with this? In the para around line 137 several experiments are mentioned but it would be useful to show simulations to compare with the data.

We agree it would be much better to compare our KO experiments with physiological experiments. We now add these comparisons to data for the SynGap KO, but we cannot find data for the effect of Epac KO. We have also added additional KO experiments, comparing the effect of Raf, Ras and Rap KO with experiments (Figure 8 – Supplementary Figure 3, lines 320-339). The effect of an Epac KO is now a prediction of the model as explained in the discussion on lines 481-486.

2. What is the relative strength of contributions by each of the proposed 5 pathways to ERK regulation? How was this ascertained?

We now present a more complete pathway analysis. Figure 8C shows the contribution of each pathway as the percentage of total activity (the ratio of ppERK AUC for a single pathway divided by ppERK AUC with all pathways), and shows that the sum of the pathways exceeds 100%. We also added additional figures (Figure 7 – Supplementary Figure 2B, Figure 6 – Supplementary Figure 1B) showing ppERK responses to single pathways, which also show that the pathways contribute to different time frames of ERK activation. We also show the dynamics of the pathways in response to single pulse stimulation in Figure 5C and D.

3. It is interesting to see differences in linearity between cAMP and Ca pathways. However, these are themselves downstream of the synaptic input. Could the authors explore how these differences would manifest upon synaptic input to a typical synapse having AMPAR, NMDAR, and mGluR? Won't they be rather overlaid on each other?

The simulations using 4 trains of both calcium and cAMP are meant to represent the response to 4 trains of synaptic activation (line 285-286). In addition, to better evaluate the response to more typical synaptic activation, we added simulations in response to the cAMP and calcium concentration dynamics that occur in response to glutamate plus neuromodulators during L-LTP induction, as determined by our published model of those pathways (Jȩdrzejewska-Szmek et al., 2017). These simulations were done by Joanna Jedrzejewska-Szmek, who has been added as a co-author. These results, new figure 9, shows that isopropanol followed by either 100 Hz or 5 Hz produce greater ppERK than ISO, 100 Hz or 5 Hz alone, which corresponds to experimental data showing that only isopropanol plus 100 Hz or 5 Hz produce L-LTP. The purpose of simulating these pathways separately is to understand their different contributions. Figure 8-supplementary Figure 2 shows that even when these pathways are correlated, they contribute to different time frames of ERK activation.

4. While it is nice to see a model with as many as 5 pathways driving ERKII, I am curious about the choice of these input pathways. From synaptic input one would expect to see an mGluR component. Do the authors envision that this folds into Gbg? Further, one sees other signals such as BDNF as major players in plasticity.4b. I bring this up also in the context of the reported time-courses. Experimental time-courses of other pathway inputs can be quite different from those explored here, BDNF tending to be quite slow.

mGluR would provide a Gq component leading to activation of PKC. This pathway is quite important for LTP in other systems (striatum, neocortex), and PKC is critical for maintenance of synaptic plasticity; however, this research is focused on induction of LTP, which is generally found to be independent of PKC. In addition, published data (Zhai et al., 2013) shows that structural plasticity, a correlate of LTP, is independent of mGluR stimulation. We now discuss this on lines 426-428.

Regarding BDNF, we agree that it contributes to ERK activation through the canonical pathways. However, as the reviewer points out, this pathway is quite slow, and thus is unlikely to contribute to temporal sensitivity. Because this research is focused on temporal sensitivity, BDNF was excluded from the model. To completely predict whether ppERK is sufficient for plasticity, ideally, we would include the BDNF-TrkB pathway; however, the concentration profile of BDNF is not well characterized, making this aspect of the model problematic.

Reviewer #2:There are, however, a couple of points that I feel should be addressed, in order for me to enthusiastically recommend this manuscript for publication.1. There needs to be additional technical detail on how the original models were expanded. The model presented here was developed by merging Jȩdrzejewska-Szmek et al., 2017 and Jain and Bhalla, 2014 models. These models were developed based on experimental data and validated with independent experimental datasets in a rigorous manner. It is not clear how the combining these two models, and the additional molecules and reactions added have affected the dynamics of ERK activation, and how comparable they are to the original experimental data used for model development in the previous modeling efforts. It is not clear if the model was reparametrized.

We now add tables of parameters which includes which parameters taken from the two models and which parameters were newly derived or re-parameterized. In addition, we provide additional validation figures (Figure 2, Figure 9, and Figure 8 – Supplementary figure 3) comparing model responses with published data.

2. Beyond the ERK activation traces, it would be useful for clarity sake to also include the simulated traces for the activation of the upstream molecules (PKA, RAS, RAP, etc). Given how additional changes have been made additional information should be provided to ensure that the contribution of each pathway is accurately represented.

We have added additional graphs showing dynamics of RasGTP (Figure 2A), active Epac (Figure 8 – Supplementary Figure 2B), active RasGRF (Figure 8 – Supplementary Figure 2C), and phosphorylated SynGap (Figure 8– Supplementary Figure 2D), to provide more insight into the pathways and demonstrate their contribution. These molecules were selected as showing contribution to different time frames of activation.

Reviewer #3:This paper is primarily about modeling the ERK pathway during the induction of synaptic plasticity. This pathway has been previously modeled, and this is cited in the paper. The main addition here is the addition of the effect of SynGap which is necessary in some form of LTP. This is a very detailed study, and what it seems to primarily show is that the ERK pathway favored spaced vs. massed stimulation protocols. This is a very detailed paper, but no conceptually new ideas are presented here. The paper adds to an existing foundation, but fails to make the case that this is a very significant addition. What is the significant consequence at a higher level of these added details?

We have completely rewritten the introduction and the discussion to emphasize the novelty of our results. Aside from demonstrating the role of SynGap, we also show that different signaling pathways contribute to different time frames of ERK activation. Thus, while ERK can respond to either spaced or massed stimulation (though we higher total activity with spaced), two of the pathways (RasGRF and Epac) are active early, SynGap is activated a bit later but prolonged, and the two other pathways are active later but transient (PKA and Gbg). These different time frames of ERK can serve allow temporal pattern discrimination. We now explain the consequence of temporal pattern discrimination in the discussion (444-492).

The ERK pathway is just one component of a much larger set of pathways that control synaptic plasticity, how much do we learn from studying this pathway in isolation? Also, the paper cites the importance of this pathway to L-LTP, is it the induction phase of L-LTP? It seems so because ppRRK decays in less than an hour. How then does this pathway contribute to the maintenance of L-LTP, these processes such as a possible upregulation of protein synthesis are not part of this model either.

We now clarify in the introduction that ERK is involved in induction of L-LTP, which includes triggering the protein synthesis (lines 59-62) The evidence for ERK in maintenance phase has not been demonstrated. In fact, very few molecules, with the exception of atypical forms of PKC, are involved in maintenance. We now explain that our time frame of ERK activation is consistent with experiments, and that ERK is not needed for maintenance (418-426).

This paper studies in detail different pathways that influence ERK activation is synapses. This is a very detailed study, but how many details do we actually know? For a detailed paper though it seems that many of the details are missing. Is there a detailed diagram of reactions, or set of equations for all these reactions? Some coefficients are named in figure 1, and this might be sufficient for a schematic description of the model in the paper, however there must be somewhere a detailed description of all reactions. How many species are there here, how many coefficients? How are coefficient values known? How many coefficients are directly estimated? The paper does carry out an extensive robustness analysis, though it is not well explained.

We now provide table of reactions and rate constants, which also indicate which coefficient values were known, and which were estimated (Figure 1 –Source Data 1-6). As is evident from these tables, very few of the coefficient values were changed.

We have re-worked the explanation of the robustness analysis (lines 366-370 and 604-609).

To summarize, the paper is very detailed carefully constructed and executed, it fails to convince that the problem it addresses is very significant, and it makes no conceptual breakthroughs.

The observation that different signaling pathways contribute to different time frames of ERK activation is a conceptual breakthrough in highlighting that, as observed in other cells, different ERK dynamics may have different functions.

The induction protocols seem to include not only calcium pulses, but also cAMP and g-protein coupled pulses, what is the cellular origin of these other pulses? Is the G protein activation due to β-adrenergic activation, is this activation necessary for plasticity? What is the cellular origin of the cAMP and Gi activation in slice? Similarly, what primarily activated cAMP in real synapses?

The cellular origin of these other pulses are Calcium, norepinephrine and dopamine. Both the locus coeruleus and the ventral tegmental area project to the hippocampus, and electrical stimulation is known to release norepinephrine, as explained in (Jȩdrzejewska-Szmek et al., 2017). Some early publications (Huang and Kandel, 1995) show that blocking dopamine will block L-LTP. Both Dopamine D1 and β-adrenergic activation produces GsGTP in the slice, which synergistically enhances the activation of adenylyl cyclase by calcium-calmodulin, to produce cAMP. I.e., in real synapses, both calcium and GsGTP produce cAMP. Regarding Gi activation, most of the research on Gs-Gi switching has been done in cell cultures (Daaka et al., 1997; Martin et al., 2004) however there is evidence that inactivated β adrenergic receptors can recruit ERK in the hippocampus (Havekes et al., 2012). The switching pathway is explained in Jȩdrzejewska-Szmek et al., 2017, and we summarize this in the methods, (lines 557-560).

Is ERK required for all forms of LTP? Some references indicate it might not be. Is ERK necessary for early LTP induced by 100Hz stimulation?

We have now clarified the difference between E-LTP and L-LTP (lines 52-58). Indeed, ERK is not required for E-LTP (Winder et al., 1999), hence we focus on induction protocols for L-LTP. We have now added simulations of E-LTP using 1 train of 100 Hz, and this shows that ppERK is much lower with these protocols.

I get lost in the details of the introduction, description of different pathways should be in results and relate to figure 1. It looks like a shopping list, why is this significant?

We have completely re-written the introduction and added more emphasis on the goal of our research, especially the importance of multiple pathway to ERK with respect to temporal pattern discrimination. We have removed many signaling pathways details and focus more on the relevant factors.

Why is it important that this is a stochastic diffusion reaction model rather than a deterministic model? Nothing about fluctuations, except for small error bars are mentioned in this paper.

When we began this study, we did not know whether stochastic fluctuations would impact our results. We tested that by doing some simulations in very small compartments, but saw no differences (except for increased noise). Though we could have switched to deterministic, the stochastic algorithm is extremely fast, especially for stiff systems; thus, there would be no advantage to switching to a potentially less accurate deterministic simulator.

Do not reference Zhang et al. Nature Neuro 2012, its' not a mammalian LTP model, still there are similarities (and differences), especially as ERK pathways are studied and since it relates to spaced learning.

We removed this article.

229 – where do we see autophosphorylation in figure 3F?

We clarified the autophosphorylation (now Figure 4F). The figure is not showing the autophosphorylation of ppERK but shows the effect of CaMKII phosphorylation of SynGap on ERK activation. We now state on lines 197-199:

“Simulations reveal that persistent activation of ERK requires phosphorylation of SynGap by pCaMKII and dispersion of pSynGap to the dendrite (Figure 4F)”

Figure 6B – hard to see shape of symbols.

We have made the size of the symbols bigger (now Figure 7B).

No model diagrams or equations here, for example how in CaMKII autophosphorylation implemented? Figure 6C – what is old CaMKII? They give a reference, but we do not know what the new model is.

We now provide a table with all reactions and rate constants. In summary, the new CamKII was created using automatic parameter optimization and fitting to time-course data of DeKoninck and Shulman, 1988. The reaction diagram also was modified slightly because NeuroRDv1 only allowed uni- and bi-molecular reactions, whereas NeuroRDv2 allowed higher-order reactions. In both cases, the higher-order phosphorylation reactions allow a single subunit model of CaMKII to exhibit a phosphorylation response of the dodecameric/hexameric in situ CaMKII. We have renamed this to alt CaMKII for alternative CaMKII model.

Narrow picture by looking only at ERK. – For example, increasing PP1 can enhance ppERK, however it might reduce overall LTP due to dephosphorylation of AMPA receptors. What wins?

We have not explained the effect of PP1 concentration clearly. Increasing PP1 does not enhance ppERK; on the contrary, it reduces ERK activation (Figure 4D), because an increase in PP1 results in less phosphorylated CaMKII. Thus, PP1 would both reduce ppERK and reduce AMPA phosphorylation. This section was meant to show that PP1 concentration changes ppERK temporal sensitivity. We have now clarified that increasing PP1 reduces total ERK activity (Figure 4D), and it decreases the steepness of the ultrasensitivity of ppERK to calcium (Figure 1E).

Figure 7 – what in figure 7 shows super linearity? It's in the title, where else? This applies to paragraph starting in line 368 as well. Where do we see these results. We can try to eyeball it. It would be useful to show a plot of combined model/(sum of individual models). By eyeballing this it seems (but might be wrong) that the super-linearity is minimal.

We now add a statement in the caption (now Figure 8) that “Supralinearity is indicated by the response to the combination of cAMP and calcium (Combo) is greater than the sum of responses to calcium and cAMP separately.” In addition, we now plot contribution of each pathway separately, and show the sum of contributions exceeds 100% of the response to the combination (Figure 8C) to help show the supralinearity. We agree that the supralinearity is not dramatic. We have also enhanced our statistical analysis (Table 5) to provide a more complete picture of linear versus non-linear pathway interactions.

403 – the first time random-forest was mentioned in line 403, and this method is not explained or defined prior to this or even in this paragraph. For eLife type of readers this needs to be explained in simple terms.

We now provide a brief explanation of random forest (lines 366-370 and 604-609).

423 – typo – "Molecules were randomly simultaneous(ly)"

Fixed

What to weights in table 5 mean? This relates to the random forest analysis which is not explained.

We now explain that the weight indicates its usefulness in predicting the dependent variable (lines 373 and 608).

Similar problems with figure 8, what is shown here? What is the analysis used here? This is totally unclear. Are molecules here changed one at a time? How much are they changed? Is this the 10% discussed earlier? It would also be more instructive to show relative range changes.

We now provide better explanation of parameter analysis and changed the presentation of the results. We now show relative change in ppERK, averaged across all ITIs, and the relative change in temporal sensitivity, quantified as the difference between maximum ppERK and minimum ppERK (across ITIs). Instead of showing the change for each parameter, we show a histogram of these values as better representing the robustness to parameter change. Using this presentation, we show that the temporal sensitivity of ppERK is quite robust, and in fact most of the parameter variations would increase, rather than decreasing temporal sensitivity.

How is this paper related to the recent Maki-Marttunen paper from the same lab which appeared recently(https://www.biorxiv.org/content/10.1101/2020.01.27.921254v1.abstract), it is not referenced here. That is a more general paper but seems to exclude ERK. If ERK is so significant how can it be excluded in one paper and included in another from the same lab?

Maki-Marttunen is studying STDP in the neocortex. There are several major differences between STDP and L-LTP. Protein synthesis dependence of STDP has not been demonstrated. This coupled with the duration of follow-up (1 hour or less due to using whole-cell patch) makes it is difficult to determine whether STDP corresponds to E-LTP or L-LTP. Regardless of whether STDP represents E-LTP or L-LTP, the requirement for ERK has not been shown. Because of the size of the synaptic plasticity field, we tried to limit our references to ERK-dependent plasticity.

Discussion starts OK, but again devolves into excessive details.

We have now re-worked the discussion, removed many details, and focus most on important aspects.

Does this model possibly explain induction of L-LTP, or its maintenance phase? It seems the prior since ppERK activity returns to baseline; these should be distinguished.

We now emphasize that the model is explaining L-LTP induction and not maintenance. (lines 59-62 and 418-426)

References:

Bhalla, U.S., 2017. Synaptic input sequence discrimination on behavioral timescales mediated by reaction-diffusion chemistry in dendrites. *eLife* 6, e25827

Daaka, Y., Luttrell, L.M., Lefkowitz, R.J., 1997. Switching of the coupling of the beta2-adrenergic receptor to different G proteins by protein kinase A. Nature 390, 88–91.

Delamater, A.R., Holland, P.C., 2008. The influence of CS-US interval on several different indices of learning in appetitive conditioning. J. Exp. Psychol. Anim. Behav. Process. 34, 202–222.

Harvey, C.D., Ehrhardt, A.G., Cellurale, C., Zhong, H., Yasuda, R., Davis, R.J., Svoboda, K., 2008a. A genetically encoded fluorescent sensor of ERK activity. Proc. Natl. Acad. Sci. U. S. A. 105, 19264–19269.

Harvey, C.D., Yasuda, R., Zhong, H., Svoboda, K., 2008b. The Spread of Ras Activity Triggered by Activation of a Single Dendritic Spine. Science 321, 136–140.

Havekes, R., Canton, D.A., Park, A.J., Huang, T., Nie, T., Day, J.P., Guercio, L.A., Grimes, Q., Luczak, V., Gelman, I.H., Baillie, G.S., Scott, J.D., Abel, T., 2012. Gravin Orchestrates Protein Kinase A and β2-Adrenergic Receptor Signaling Critical for Synaptic Plasticity and Memory. J. Neurosci. 32, 18137–18149.

Huang, Y.Y., Kandel, E.R., 1995. D1/D5 receptor agonists induce a protein synthesis-dependent late potentiation in the CA1 region of the hippocampus. Proc. Natl. Acad. Sci. U. S. A. 92, 2446–2450.

Jȩdrzejewska-Szmek, J., Luczak, V., Abel, T., Blackwell, K.T., 2017. β-adrenergic signaling broadly contributes to LTP induction. PLOS Comput. Biol. 13, e1005657.

Kasahara, J., Fukunaga, K., Miyamoto, E., 2001. Activation of Calcium/Calmodulin-dependent Protein Kinase IV in Long Term Potentiation in the Rat Hippocampal CA1 Region*. J. Biol. Chem. 276, 24044–24050.

Keyes, J., Ganesan, A., Molinar-Inglis, O., Hamidzadeh, A., Zhang, Jinfan, Ling, M., Trejo, J., Levchenko, A., Zhang, Jin, 2020. Signaling diversity enabled by Rap1-regulated plasma membrane ERK with distinct temporal dynamics. *eLife* 9, e57410.

Martin, N.P., Whalen, E.J., Zamah, M.A., Pierce, K.L., Lefkowitz, R.J., 2004. PKA-mediated phosphorylation of the β1-adrenergic receptor promotes Gs/Gi switching. Cell. Signal. 16, 1397–1403.

Mauk, M.D., Ruiz, B.P., 1992. Learning-dependent timing of Pavlovian eyelid responses: differential conditioning using multiple interstimulus intervals. Behav. Neurosci. 106, 666–681.

McKeever, P.M., Kim, T., Hesketh, A.R., MacNair, L., Miletic, D., Favrin, G., Oliver, S.G., Zhang, Z., St George-Hyslop, P., Robertson, J., 2017. Cholinergic neuron gene expression differences captured by translational profiling in a mouse model of Alzheimer’s disease. Neurobiol. Aging 57, 104–119.

Winder, D.G., Martin, K.C., Muzzio, I.A., Rohrer, D., Chruscinski, A., Kobilka, B., Kandel, E.R., 1999. ERK Plays a Regulatory Role in Induction of LTP by Theta Frequency Stimulation and Its Modulation by β-Adrenergic Receptors. Neuron 24, 715–726.

Zhai, S., Ark, E.D., Parra-Bueno, P., Yasuda, R., 2013. Long-Distance Integration of Nuclear ERK Signaling Triggered by Activation of a Few Dendritic Spines. Science 342, 1107–1111.

Zhang, L., Zhang, P., Wang, G., Zhang, H., Zhang, Y., Yu, Y., Zhang, M., Xiao, J., Crespo, P., Hell, J.W., Lin, L., Huganir, R.L., Zhu, J.J., 2018. Ras and Rap Signal Bidirectional Synaptic Plasticity via Distinct Subcellular Microdomains. Neuron 98, 783–800.

[Editors' note: further revisions were suggested prior to acceptance, as described below.]

Point 1: Comparisons to experiments and validation of parameterization.The authors have partly addressed this point, with comparisons of simulations to several additional experiments. They have also provided tables of data sources but many parameters are linked to previous models rather than to experiments. It would be helpful if the authors could explicitly address the point about _parameter_ validation, and to what extent systems level experiments provide validation for rate terms.

Parameter validation is indeed a critical, yet difficult task for signaling pathway models. We have updated out tables to include the experimental data sources used in the previous models, to show that many of the parameters were constrained by experimental data. In addition, we provide supplementary figures (Figure 2- supplementary Figure1) showing the fit of the model to data for CaMKII and SynGap (which are the results of parameter optimization), and cAMP-bound Epac and phosphorylation of PKA substrates (which are systems level validations). Since publication of time course data is not common, we are unable to provide direct validation of additional parameters. As the tables illustrate, most of the parameters come from well-validated models, in which parameters were constrained experimentally (e.g. Sasagawa et al., Nature Cell Biology, 2005) or which made predictions that were experimentally tested. The fact that those published parameters are sufficient to reproduce phosphoERK from additional published experiments (Figure 2B) provides yet another systems level validation for the parameters in the published models. Systems level validation is common with these types of models in part due to the difficulty in accurately measuring on rates (compared to affinity) using biochemical experiments. We have added additional sentences about the parameter validation on 116-123.

Point 2: Activation of pathways by synaptic input.Here the authors use the model from their 2017 paper to generate Ca and cAMP waveforms, and report results consistent with experiments. It is just a single figure though, and only reports ppERK. We don't see the upstream pathways and their responses. Since the authors have now added these upstream pathways, it would be good to see the time-courses for upstream key molecules indicated in figure 1. These include Ca, Gbg, cAMP, as well as CaMKII, PKA, synGap, Ras and Rap1. It would be useful to have a figure with these time-courses for the major stimulus patterns used in the figures. In Figure 8 – supplementary Figure 2B the authors provide some time-courses for a few molecules. A complete set would be desirable.

We now provide supplementary figures showing the time course of calcium, cAMP and Giβγ for single pulse (line 147, 182-183 and Figure 3- supplementary Figure1). We also provide time course of calcium, Giβγ, cAMP, CaMKII, PKA, synGap, Ras, Rap1, Raf, MEK, pSrc in Figure 8 – supplementary Figure 1 and Figure 9 – supplementary Figure1.

Point 5: Source data now provided. Unfortunately it isn't in SBML. If the authors can apply a conversion program to their model, an SBML version would be valuable.

A neurord XML to SBML program has recently been written, though it does not yet convert a small subset of reactions correctly and it does not convert the stimulation files. We now include the SBML for the main reaction and initial condition file in our github repository, and this will be SBML file will be updated once the conversion program has been debugged.

Also, the response to the reviewer's point about the use of a stochastic method would be good to include in the text. We did not see this statement there.

We now include a statement in the methods (lines 593-596) that “Even though stochastic fluctuations observed using small compartments do not impact the results, the stochastic algorithm is extremely fast, especially for stiff systems; thus, there would be no advantage to switching to a potentially less accurate deterministic simulator.”